# Evaluating distributional regression strategies for modelling self-reported sexual age-mixing

Timothy M Wolock[1]*, Seth Flaxman[1], Kathryn A Risher[2,3], Tawanda Dadirai[4], Simon Gregson[2,4], Jeffrey W Eaton[2]

[1]Department of Mathematics, Imperial College London, London, United Kingdom; [2]MRC Centre for Global Infectious Disease Analysis, School of Public Health, Imperial College London, London, United Kingdom; [3]London School of Hygiene & Tropical Medicine, London, United Kingdom; [4]Manicaland Centre for Public Health Research, Biomedical Research and Training Institute, Harare, Zimbabwe

**Abstract** The age dynamics of sexual partnership formation determine patterns of sexually transmitted disease transmission and have long been a focus of researchers studying human immunodeficiency virus. Data on self-reported sexual partner age distributions are available from a variety of sources. We sought to explore statistical models that accurately predict the distribution of sexual partner ages over age and sex. We identified which probability distributions and outcome specifications best captured variation in partner age and quantified the benefits of modelling these data using distributional regression. We found that distributional regression with a sinh-arcsinh distribution replicated observed partner age distributions most accurately across three geographically diverse data sets. This framework can be extended with well-known hierarchical modelling tools and can help improve estimates of sexual age-mixing dynamics.

*For correspondence:
t.wolock18@imperial.ac.uk

Competing interests: The authors declare that no competing interests exist.

## Introduction

Patterns in sexual mixing across ages determine patterns of transmission of sexually transmitted infections (STIs). Consequently, sexual age-mixing has been of great interest to researchers studying the human immunodeficiency virus (HIV) since the beginning of the global epidemic. *Anderson et al., 1992* used a model of partnership formation to predict that mixing between young women and older men would amplify the already-substantial effect of HIV on population growth. *Garnett and Anderson, 1994* used a mathematical model to show that patterns of age-mixing could substantially influence the magnitude and timing of hypothetical epidemic trajectories, whereas *Hallett et al., 2007* demonstrated that delaying sexual debut and increasing age-similar partnerships could reduce an individual's risk of HIV infection in a highly endemic setting.

These modelling studies have been complemented by analyses of survey and population cohort data on age-mixing patterns. *Gregson et al., 2002* and *Schaefer et al., 2017* observed that individuals with older partners were at greater risk of HIV infection in a general-population cohort in Zimbabwe. *Ritchwood et al., 2016* and *Maughan-Brown et al., 2016* found that larger age differences were associated with more risky sexual behaviour in surveys of young South African people. Similarly, *Akullian et al., 2017* found that partner age was an important risk factor for HIV infection in a cohort study in rural South Africa. On the other hand, *Harling et al., 2014* found that age-disparate relationships were not associated greater risk of HIV acquisition in young women in South Africa.

These results underscore the importance of considering age-mixing dynamics when designing and evaluating HIV prevention strategies, and, consequently, the importance of measuring them accurately. For example, an intervention aiming to prevent new HIV infections among young women

could be attenuated by high prevalence among older men. Identifying changes in sexual partner age distributions and attributing them to interventions might even be a valuable end by itself, in which case accurate measurement must be complemented by an effective modelling strategy.

Data about sexual partner age-mixing are routinely collected by long-term cohort studies (such as those that comprise the Analysing Longitudinal Population-based HIV/AIDS data on Africa, or ALPHA, Network) and large-scale household surveys (such as the Demographic and Health Surveys) (*The DHS program, 2021*; *Reniers et al., 2016*). Typically, these data consist of the respondent's age and sex and the ages of their sexual partners in the last 12 months. These data are highly variable, skewed, and often deviate substantially from conventional parametric distributions, such as the normal distribution or the gamma distribution (*Beauclair et al., 2018*).

One may consider statistical modelling approaches for the distribution of partner age as a function of respondent age and sex. Some notable previous approaches to modelling partner age distributions include (*Morris, 1991*), who developed a log-linear modelling framework to quantify selective mixing from contingency tables, and applied the model to ordinal categorical data on husbands' and wives' ages in the United States. Hallett et al. used a log-logistic distribution to continuously model partner age differences for women aged 15–45 years, assuming that the partner age difference distributions did not vary over respondent age. More recently, as an input to a model of *Chlamydia trachomatis*, *Smid et al., 2018* fit skew normal distributions to each age-/sex-specific partner age distribution and used a secondary regression model to smooth the estimated skew normal parameters across respondent age. They observed substantial changes in the estimated skew normal parameters with respect to respondent age. Although this method allows for non-linear variation across respondent age, their two-stage estimation process makes uncertainty propagation complex. Replacing this process with a single 'distributional' regression model, in which all distributional parameters (e.g. the location, scale, skewness, etc.) are modelled as functions of data (*Kneib and Umlauf, 2017*), allows for complex variation across respondent age while still robustly incorporating uncertainty. Another elegant approach has been the development of exponential-family random graph models (ERGMs) to infer full partnership networks from individuals reports of the partnerships ('ego-centric' observations of the network) (*Krivitsky and Morris, 2017*). These stochastic methods, along with the broader suite of ERGMs (*Hunter et al., 2008a*; *Hunter et al., 2008b*; *Krivitsky et al., 2011*; *Krivitsky and Handcock, 2014*), can model social network data accurately with robust incorporation of covariates, and tools exist to incorporate their estimates into epidemic models (*Jenness et al., 2018*; *Morris, 1993*).

We focused on evaluating parametric models for continuously representing the distribution of sexual partner ages conditional on the respondent's age. We were specifically interested in distributions that introduce parameters to control tail weight, which may capture intergenerational mixing that could sustain endemic HIV and STI transmission (*Akullian et al., 2017*; *Schaefer et al., 2017*; *Harling et al., 2014*). This led us to test the ability of the four-parameter 'sinh-arcsinh' distribution originally proposed by *Jones and Pewsey, 2009* to fit to these data.

We hypothesised that integrating the sinh-arcsinh distribution into a distributional modelling framework would allow us to replicate observed partner age distributions more accurately than prior modelling strategies. We tested this theory by comparing a variety of candidate strategies, which varied along three dimensions: the parametrisation of the dependent variable, the choice of distribution, and the method for incorporating variability across respondent age and sex.

## Materials and methods

### Data

In this work, we compared a set of probability distributions and regression specifications to identify a modelling strategy that produced stable and accurate estimates of sexual partner age distributions with well-quantified uncertainty. We conducted two model comparison experiments to identify which of a set of strategies best replicated partner age distributions. First, in our probability distribution comparison, we identified which of a set of distribution-dependent variable combinations fit best to age-/sex-specific data subsets, and then, in our distributional regression evaluation, we tested whether distributional regression methods could be used to estimate age-/sex-specific partner age distributions by sharing strength across observations. In distributional regression, all

parameters of a distribution can vary with respect to data, so it allowed us to smooth and interpolate even higher order moments of observed partner ages. We divided the model comparison into two separate experiments to make the probability distribution comparison as fair as possible (accounting for the possibility that certain distributions would perform particularly well under certain regression specifications).

We analysed data on sexual partner age distributions from three sources: the Africa Centre Demographic Information System, a health and demographic surveillance site in uMkhanyakude district, South Africa collected by the African Health Research Institute (AHRI) (*Gareta et al., 2021*; *Gareta et al., 2020a*; *Gareta et al., 2020b*), the Manicaland General Population Cohort in Zimbabwe (*Gregson et al., 2017*), and the 2016–2017 Demographic and Health Survey (DHS) in Haiti (*Institut Haïtien de l'Enfance, 2018*).

The AHRI and Manicaland studies are multi-round, open, general population cohort studies designed to measure the dynamics of HIV, sexual risk behaviour, and demographic change in sub-Saharan African settings. We used rounds one through six of the Manicaland study, collected between 1998 and 2013. The AHRI data we used were collected annually between 2004 and 2018. The 2016–17 Haiti DHS was a large, nationally representative household survey conducted in 2016 and 2017. We did not incorporate the weights associated with the survey into this analysis because our primary interest was in statistical modelling of partner age distribution as a function of respondent age, not producing population representative statistics for the Haitian population.

These data sets consisted of individuals' reports of their own age and sex and the ages of each of their sexual partners from the last year. Let $i \in (1, ..., N)$ index reported partnerships, $a_i \in [15, 64]$ and $s_i \in \{0, 1\}$ be the age and sex of the respondent in partnership $i$ with $s = 1$ indicating female, and $p_i$ be the age of non-respondent partner in partnership $i$. These questionnaires do not ask specifically about partner sex, but self-reporting of non-heterosexual partnerships in these populations is thought to be low (*Arias Garcia et al., 2020*; *World Health Organization and UNAIDS, 2020*).

Respondents in each of these data sets are disproportionately likely to report that their partners' ages are multiples of five or multiples of five away from their own age, leading to distinct 'heaping' in the empirical partner age (or age difference) distributions at multiples of five. For example, if a questionnaire asks 'how many years older or younger is your partner than you?', respondents might be disproportionately likely to report a multiple of five, leading to age *differences* that are heaped on multiples of five. We tested the sensitivity of our results to heaping by developing a simple 'deheaping' algorithm, applying it to the AHRI data, and running each analysis on the deheaped AHRI data. We present these results in Appendix section 'Age heaping'.

## Probability distribution comparison

To identify the best probability distribution for modelling sexual partner age distributions, we split all three data sets into 12 subsets by sex and five-year age bin ranging from 20 to 50, resulting in 36 subsets, and fit a number of distribution-dependent variable combinations to each subset.

### Distributions

We tested five candidate probability distributions: normal, skew normal, beta, gamma, and sinh-arcsinh. *Table 1* summarises the domains, parameters, and probability density functions (PDFs) of these distributions. Because the gamma distribution is always right-skewed and men typically partner with women who are younger than them, we transformed data among male respondents to be right-skewed when using the gamma distribution. Specifically, we multiplied the men's partners' ages by −1 to reflect the distribution horizontally across the y-axis, and added 150 to the reflected ages to ensure that all resulting values were positive. Similarly, the beta distribution is only defined on the interval (0, 1), so, only when using a beta distribution, we scaled all partner ages to be between zero and one using upper and lower bounds of 0 and 150.

The sinh-arcsinh distribution, presented by *Jones and Pewsey, 2009*, is an extension of Johnson's distribution (*Johnson, 1949*). It has four parameters: location, scale, skewness, and tail weight (denoted, $\mu$, $\sigma$, $\epsilon$, and $\delta$, respectively), and it can deviate substantially from the normal distribution. *Figure 1* plots the density of this distribution with $\mu = 0$ and $\sigma = 1$ for a variety of values of skewness and tail weight.

**Table 1.** Details of the five distributions tested in this analysis.

We define $x_z = (x - \mu)/\sigma$, $p(x)$ to be the standard normal PDF, $\Phi(x)$ to be the standard normal cumulative density function, $S_{\epsilon,\delta}(x) = \sinh(\epsilon + \delta \operatorname{asinh}(x))$, and $C_{\epsilon,\delta}(x) = \cosh(\epsilon + \delta \operatorname{asinh}(x))$.

| Distribution | Parameters | Domain | PDF |
|---|---|---|---|
| Normal | $\mu$ (location)<br>$\sigma > 0$ (scale) | $\mathbb{R}$ | $\frac{1}{\sigma\sqrt{2\pi}} \exp\left[\frac{-x_z}{2}\right]$ |
| Skew normal | $\mu$ (location)<br>$\sigma > 0$ (scale)<br>$\epsilon$ (skewness) | $\mathbb{R}$ | $\frac{2}{\sigma} p(x_z) \Phi(\epsilon x_z)$ |
| Gamma | $k > 0$ (shape)<br>$\theta > 0$ (scale) | $\mathbb{R}^+$ | $\frac{1}{\Gamma(k)\theta^k} x^{k-1} \exp\left[\frac{-x}{\theta}\right]$ |
| Beta | $\alpha > 0$ (left)<br>$\beta > 0$ (right) | $\mathbb{R}^{(0,1)}$ | $\frac{x^{\alpha-1}(1-x)^{\beta-1}}{B(\alpha,\beta)}$ |
| Sinh-arcinh | $\mu$ (location)<br>$\sigma > 0$ (scale)<br>$\epsilon$ (skewness)<br>$\delta > 0$ (tailweight) | $\mathbb{R}$ | $\frac{1}{\sigma\sqrt{2\pi}} \frac{\delta C_{\epsilon,\delta}(x_z)}{\sqrt{1+x_z^2}} \exp\left[-\frac{S_{\epsilon,\delta}(x_z)^2}{2}\right]$ |

## Dependent variable transformations

We considered the possibility that certain distributions could fit better to particular transformations of the dependent variable (partner age) by testing a set of four potential outcome parametrisations. For example, if $X$ is a positive-valued, right-skewed random variable, then assuming $\log X$ is normally distributed might be more effective than assuming that $X$ itself is normal.

Let $y_i$ be the dependent variable value for partnership $i$, and let $a_i$ and $p_i$ be the respondent age and partner age of partnership $i$, respectively. We tested the following dependent variables:

1. Linear age: $y_i = p_i$. This is untransformed partner age, included as a baseline. It has the undesirable quality of being able to predict negative ages.
2. Age difference: $y_i = p_i - a_i$. If changes in expected partner age are consistent across respondent age then this variable would be more consistent across respondent age than the linear age. This parametrisation also allows for negative partner age predictions.
3. Log-age: $y_i = \log p_i$. We can use a $\log$ link function to ensure that our predictions will be positive-valued.
4. Log-ratio: $y_i = \log(p_i/a_i)$. Finally, we can combine the link function and differencing approaches by modelling the log of the ratio of partner to respondent age. This variable will only produce

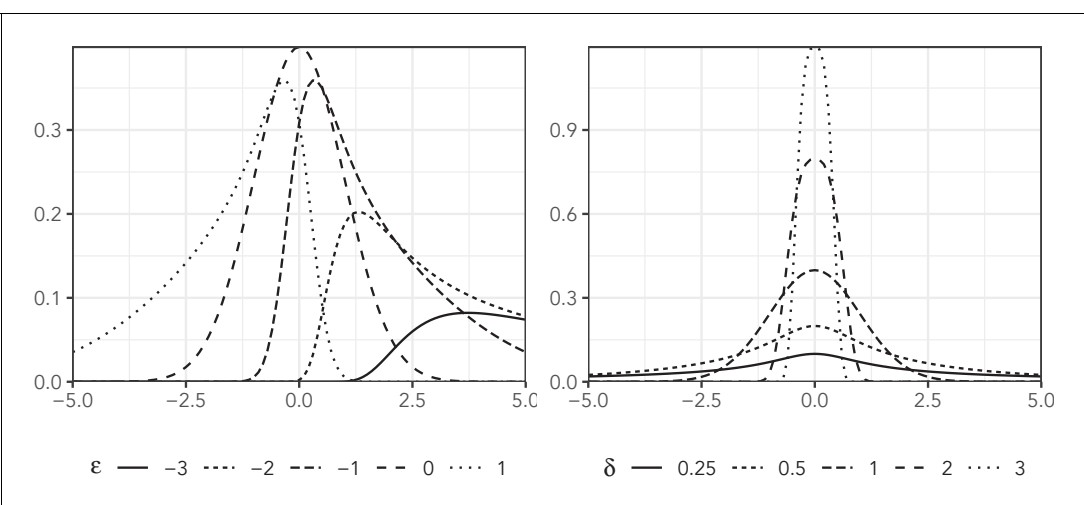

**Figure 1.** The sinh-arcsinh density with $\mu = 0$, $\sigma = 1$, and a variety of assumptions about $\epsilon$ and $\delta$.

positive predictions and, like the age difference variable, should be relatively constant over respondent age.

Because the gamma and beta distributions are not defined on the entire real line, we only fit them with the linear age dependent variable with the previously discussed transformations.

To identify which distribution-dependent variable combination best modelled the characteristics of sexual partner age distributions, we stratified each of our three data sets by sex and 5-year age bin from 20 to 24 through 45–49. We omitted ages 15–19 from the probability distribution comparison because relatively small sample sizes in that group would make reliable comparison difficult. We fit every viable distribution-dependent variable combination to all 36 data set-/sex-/age bin-specific subsets independently. Given that we fit only the linear age dependent variable to the gamma and beta distributions, comprising a total of 504 models (14 per data subset). We fit each model using the brms R package (*Bürkner, 2018*), defining custom families as necessary.

## Distributional regression evaluation

Given a probability distribution that accurately replicated the non-Gaussian characteristics of partner age distributions, we tested whether or not distributional regression would allow us to pool data across age and sex without sacrificing fit. In distributional regression, we make all of our distributional parameters, not just the mean, functions of data (*Kneib and Umlauf, 2017*). Taking conventional Bayesian regression as an example, we have

$$
\begin{aligned}
y_i &\sim \mathrm{N}(\mu_i, \sigma) \\
\mu_i &= \beta \mathbf{X}_i,
\end{aligned}
$$

where $\beta$ and $\log \sigma$ are free parameters. There is an explicit assumption in this model that the standard deviation of the generating distribution is constant across all observations. We can use distributional regression to relax this assumption, making $\sigma$ a function of data:

$$
\begin{aligned}
y_i &\sim \mathrm{N}(\mu_i, \sigma_i) \\
\mu_i &= \beta^\mu \mathbf{X}_i^\mu \\
\log \sigma_i &= \beta^\sigma \mathbf{X}_i^\sigma,
\end{aligned}
$$

where $\beta^\mu$ and $\beta^\sigma$ are now our free parameters. Note that we have not assumed that $\mathbf{X}^\mu = \mathbf{X}^\sigma$. If $\mathbf{X}^\sigma$ is a column of ones, this model is identical to the conventional case. This approach increases the complexity of the model and requires more data, but, based on previously described characteristics of how the distribution of partnership age distribution changes with age, even a simple model for our distributional parameters could yield large improvements.

In this application, we modelled the log-ratio dependent variable with the sinh-arcsinh distribution, specifying a model for all four distributional parameters:

$$
\begin{aligned}
\log(p_i/a_i) &\sim \sinh(\mu_i, \sigma_i, \epsilon_i, \delta_i) \\
\mu_i &= \beta^\mu \mathbf{X}_i^\mu \\
\log \sigma_i^\star &= \beta^\sigma \mathbf{X}_i^\sigma \\
\epsilon_i &= \beta^\epsilon \mathbf{X}_i^\epsilon \\
\log \delta_i &= \beta^\delta \mathbf{X}_i^\delta \\
\sigma_i &= \sigma_i^\star \delta_i,
\end{aligned}
$$

where $\beta^\mu$, $\beta^\sigma$, $\beta^\epsilon$, and $\beta^\delta$ are free parameters. We placed essentially arbitrary shrinkage priors on all coefficients:

$$
\beta^\mu, \beta^\sigma, \beta^\epsilon, \beta^\delta \sim \mathrm{N}(0, 5).
$$

By varying the specifications of the four design matrices, $\mathbf{X}^\mu$, $\mathbf{X}^\sigma$, $\mathbf{X}^\epsilon$, and $\mathbf{X}^\delta$, we tested how well a series of increasingly complex distributional regression models fit to each data set. We fit the following models, which varied in the definitions of their four design matrices:

1. Conventional: linear age-sex interaction for location and constants for all three higher-order parameters
2. Distributional 1: linear age-sex interaction for location and independent age and sex effects for all other parameters

3. Distributional 2: linear age-sex interactions for all four parameters
4. Distributional 3: sex-specific spline with respect to age for location and linear age-sex interactions for all other parameters
5. Distributional 4: sex-specific splines with respect to age for all four parameters

*Table 2* describes all five models. By fitting a wide set of specifications, we hoped to assess whether the additional complexity incurred by distributional regression was valuable. We fit each of the five models to all three data sets, including all respondents aged 15–64 years. We implemented these analyses with brms (*Bürkner, 2018*), which has deep support for distributional regression. More detailed descriptions of each model are available in the 'Model specification details' section of the Appendix. We tested the effect of age heaping on this analysis by fitting to the deheaped AHRI data and report results in the 'Age heaping' section of the Appendix.

## Model comparison

Across both analyses, we used two metrics to measure model fit. First, we calculated the expected log posterior density (ELPD), which estimates the density of the model at a new, unobserved data point (*Vehtari et al., 2017*). In cases where we wanted to compare across dependent variables, we multiplied the posterior densities of any variables resulting from non-linear transformations of observed partner ages by the Jacobians of the transformations. For example, if our observation model was defined on the log-age dependent variable $y_i = \log p_i$, we divided the posterior density by $p_i$. We used the loo R package (*Vehtari et al., 2020*) to calculate ELPD values.

To measure the ability of our models to replicate partner age distributions in an objective and interpretable way, we found the root mean squared error (RMSE) between the observed and posterior predictive quantiles. We calculated quantiles from 10 to 90 in increments of 10 by age bin and sex in the data and in the posterior predictions, and found the error in model prediction of each quantile. This measure tells how well our model predicts the entire distribution in the same units as our predictions. It is equivalent to finding the root mean squared distance from the line of equality in a quantile-quantile (QQ) plot.

## Software

We conducted all of these analysis using the R programming language (*R Development Core Team, 2020*) and the brms library (*Bürkner, 2018*). We used the loo library to estimate all ELPDs (*Vehtari et al., 2020*), and produced all plots in this paper with the ggplot2 library (*Wickham, 2016*). We cannot provide the data we used for this analysis, but we do provide code and data for a simulated case on GitHub (https://github.com/twolock/distreg-illustration; *Wolock, 2021*, copy archived at swh:1:rev:a7f808f2cde2bb16edde8fdcbfa6e208df7952f9).

## Results

The AHRI data included 77,619 partnerships, Manicaland had 58,676, and the Haiti DHS had 12,447. There were 36,033 respondents reporting at least one partnership in the AHRI data, 25,024 in the Manicaland data, and 12,143 in the Haiti DHS, resulting in averages of 2.2, 2.3, and 1.0 partners per respondent, respectively. As an illustrative example of the distribution of partner ages, *Figure 2* presents histograms of reported partner ages among women aged 34 years for each of our three data sets. *Figure 3* shows the sex- and age bin-specific empirical moments for the three data sets. Mean partner age increased with respondent age consistently for both sexes across all three data

**Table 2.** Summary of five models fit in this analysis.

| Model | Distributional? | Location | Other parameters |
|---|---|---|---|
| Conventional | No | Age-sex interaction | Constant |
| Distributional 1 | Yes | Age-sex interaction | Age and sex effects |
| Distributional 2 | Yes | Age-sex interaction | Age-sex interaction |
| Distributional 3 | Yes | Sex-specific splines | Age-sex interaction |
| Distributional 4 | Yes | Sex-specific splines | Sex-specific splines |

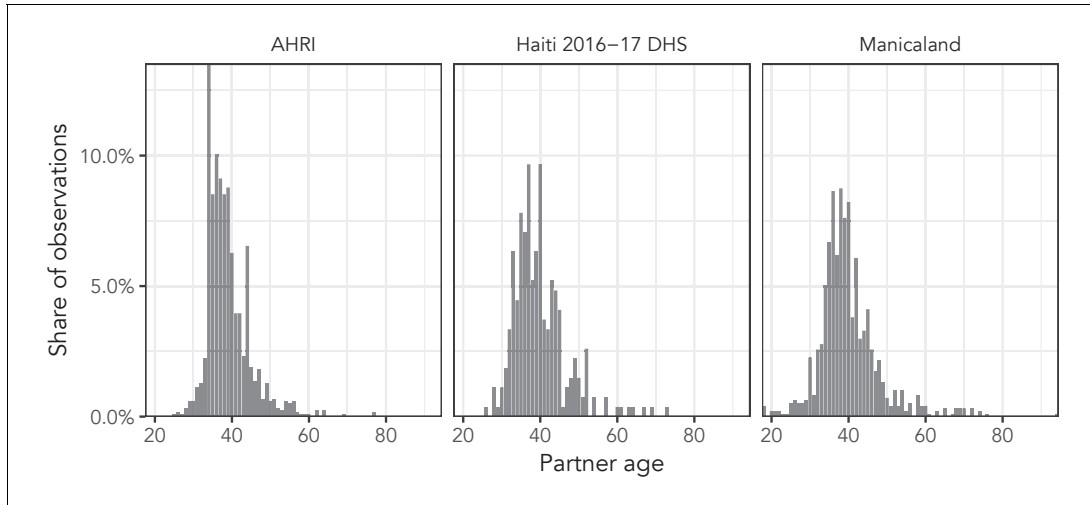

**Figure 2.** Observed partner age distributions among women aged 34 years in all three data sets.

sets: among women, mean partner age increased by 26.0, 22.7, and 23.7 years in the AHRI data, Haiti DHS data, and Manicaland data, respectively, between age bins 20–24 and 45–49. However, higher order moments were less consistent: the standard deviation of women's partners' ages changed by 2.3, 0.5, and 3.5 years in the AHRI data, Haiti DHS data, and Manicaland data, respectively, between age bins 20–24 and 45–49.

Within each data set, there is systematic variation across sex. For example, the standard deviation of partner ages in the Haiti DHS increased by 2.5 years among men and only by 0.5 years among women. These summary statistics illustrate the heterogeneity of partner age distributions across age and sex.

## Probability distribution comparison

To identify the probability distribution that most accurately described the variation in sexual partner age distributions, we first determined the dependent variable with the highest ELPD for each distribution-dependent variable combination. *Figure 4* illustrates each probability distribution's best fit to AHRI data among women aged 35–39 with each of the best distribution-specific dependent variables. Results for all 36 data subsets and the 12 deheaped subsets are presented in *Appendix 1— table 2–9*.

The best dependent variable varied across data subset and probability distribution. *Table 3* provides the share of data sets for which each dependent variable has the highest ELPD given each distribution. The log-ratiodependent variable was best in 50.0% of subsets with a normal distribution, but it was best in only 27.8% of subsets with a skew normal distribution. The dependent variable that was best in a plurality of subsets in each probability distribution (i.e. the variable with the highest percentage in each column of *Table 3*) used a $\log$ link function. We restricted all remaining comparisons to each distribution-subset combination's best dependent variable.

The sinh-arcsinh distribution had the highest ELPD in 35 of 36 data subsets (98%). In 29 of the 35 (83%) cases in which the sinh-arcsinh provided the highest ELPD, the absolute value of the ratio of the difference between the two best ELPDs and the estimated standard error of the difference was greater than 2, indicating that the sinh-arcsinh distribution was significantly better than the alternatives in the majority of cases. In one case, men aged 20–24 in the Haiti DHS, the skew normal distribution resulted in a slightly higher ELPD than the sinh-arcsinh distribution, but the standard error of the difference was greater than the difference. These results were not affected by deheaping the data (Appendix section 'Age heaping').

To summarise each distribution's performance, we calculated the average ELPD and QQ RMSE across the three data sets (*Table 4*). The sinh-arcsinh distribution had the highest average ELPD and lowest average QQ RMSE in all three data sets. The sinh-arcsinh distribution was, on average, able to predict the empirical quantiles of each data set within half a year of accuracy (0.36, 0.37, and 0.44

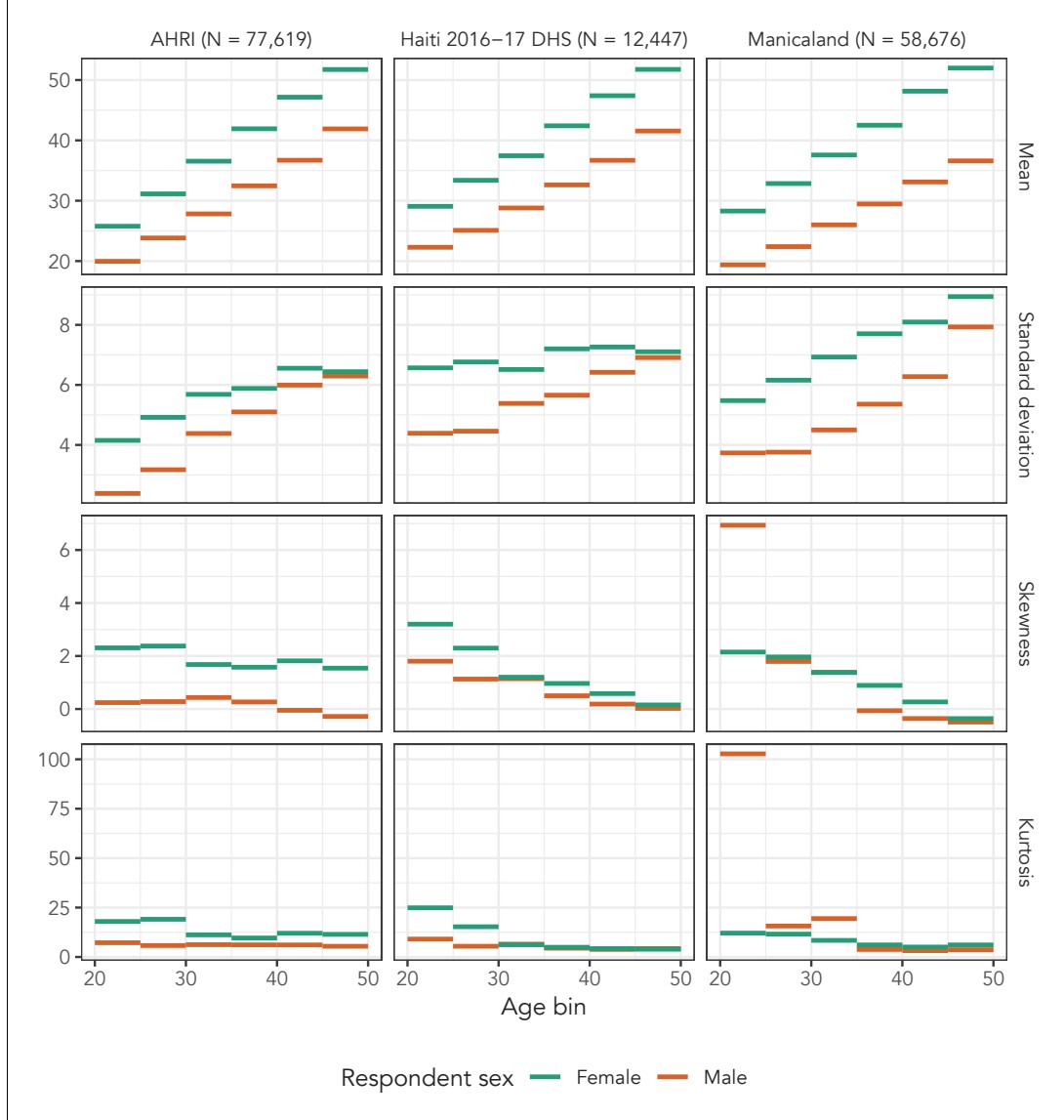

**Figure 3.** Observed means, variances, skewnesses, and kurtoses of partner age by 5-year age bin and sex in all three data sets.

years for the AHRI, Haiti DHS, and Manicaland data, respectively). Overlaid QQ plots are presented in *Appendix 1—figure 3*.

## Distributional regression evaluation

We fit all five distributional regression specifications to all three of our data sets with sinh-arcsinh distributions and log-ratio-dependent variables and compared the ELPDs and QQ RMSEs as before (provided in *Table 5*). Across all three data sets, the most complex distributional model (Distributional 4) had the highest ELPD and lowest QQ RMSE. When fit to the AHRI and Manicaland data sets (but not for the Haiti DHS), the most complex distributional model was at least two standard errors better than the next best model. Notably, the largest ELPD improvements came from moving from conventional regression (Conventional) to the simplest distributional model (improvements of 1646.0 units, 361.0 units, and 2181.2 units in the AHRI, Haiti DHS, and Manicaland data, respectively). Full results are presented in *Appendix 1—table 10*.

*Figure 5* shows the posterior predictive distributions from the conventional regression model and the most complex distributional model among men aged 16 years, 24 years, and 37 years in the AHRI data to illustrate the effect of distributional regression. Not only does the distributional model

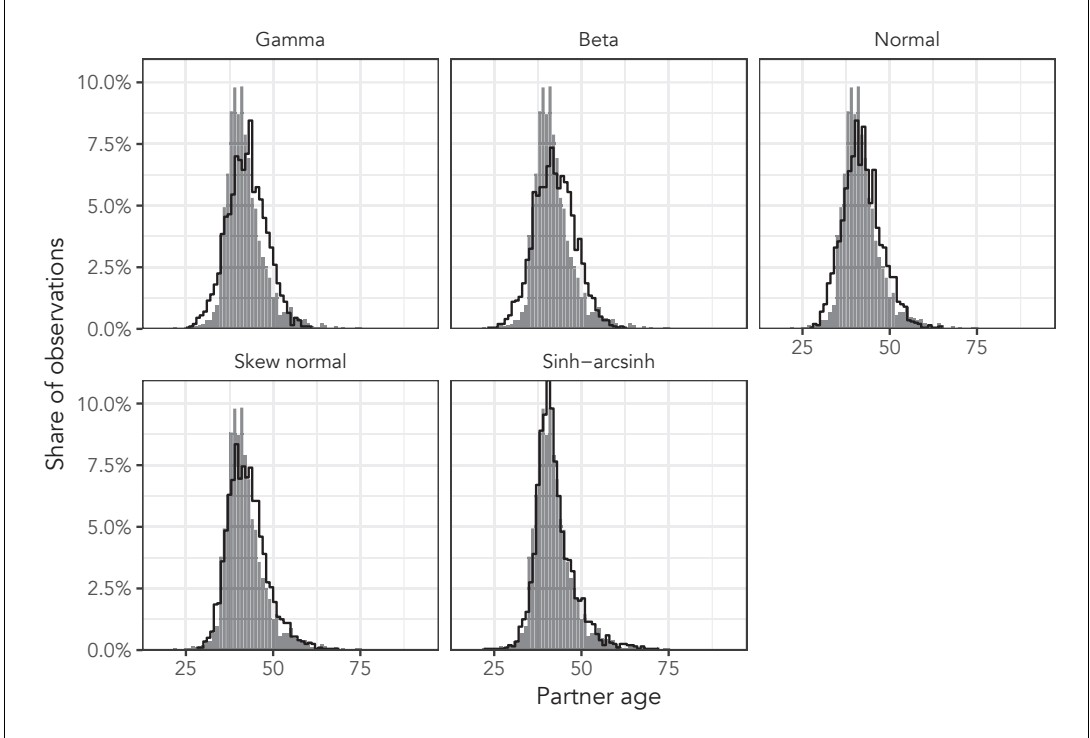

**Figure 4.** Observed partner age distributions (grey bars) and posterior predictive partner age distributions (lines) for each probability distribution among women aged 35–39 in the AHRI data set. Posterior predictive distributions come from fitting each age bin/sex combination independently.

capture the high peak in the youngest age more accurately, but it also allows the variance of the distributions to change appropriately (beyond the change that naturally results from the log link function).

*Figure 6* illustrates posterior summaries among men and women in the AHRI data for all four distributional parameters for the conventional regression model, the simplest distributional model, and the most complex distributional model. The red estimates (Conventional Regression) of the three higher order parameters were constant across age and sex, whereas the blue estimates (Distributional Model 1) included independent, linear age and sex effects. The orange estimates (Distributional Model 4) were generated sex-specific splines with respect to age, allowing for flexible variation across age and sex.

The third row of plots in *Figure 6*, which corresponds to the skewness parameter, illustrates the impact of incorporating sex and age effects into the model. The conventional regression model estimated that neither the distrbution for men nor women exhibited much skewness; the estimated parameter value was −0.05 (95% UI: −0.06 to −0.05) regardless of age, with 0.0 corresponding to perfect symmetry. However, when we allowed independent age and sex effects in Distributional Model 1, we estimated that at age 15, women's skewness was −0.26 (95% UI: −0.27 to −0.25) and men's was 0.11 (95% UI: 0.10 to 0.12).

**Table 3.** Share of subsets in which each dependent variable yields the highest ELPD given each probability distribution (excluding deheaped AHRI data).

| Variable | Normal | Skew normal | Sinh-arcsinh |
|---|---|---|---|
| Age difference | 22.2% | 25.0% | 16.7% |
| Linear age | 8.3% | 5.6% | 16.7% |
| Log-age | 19.4% | 41.7% | 30.6% |
| Log-ratio | 50.0% | 27.8% | 36.1% |

**Table 4.** Model comparison metrics averaged across all data subsets for all three data sets.
Higher ELPD values indicate better fit. Lower QQ RMSE values indicate more accurate prediction of empirical quantiles. Bolded rows are best across all three data sets.

| Distribution | AHRI | Haiti 2016–17 DHS | Manicaland |
|---|---|---|---|
| ELPD | | | |
| Gamma | −14847.2 | −2917.9 | −13152.8 |
| Beta | −14748.0 | −2896.5 | −13003.5 |
| Normal | −14593.7 | −2868.4 | −12856.8 |
| Skew normal | −14505.1 | −2854.0 | −12778.5 |
| Sinh-arcsinh | **−14312.5** | **−2839.5** | **−12625.8** |
| QQ RMSE | | | |
| Gamma | 0.83 | 0.82 | 0.95 |
| Beta | 0.99 | 0.82 | 1.11 |
| Normal | 0.82 | 0.68 | 0.97 |
| Skew normal | 0.77 | 0.65 | 0.85 |
| Sinh-arcsinh | **0.36** | **0.37** | **0.44** |

The most complex model (Distributional Model 4) inferred sex-specific, non-linear variation with respect to age in all four distributional parameters. The non-linearity was particularly dramatic in the scale parameter among men. The scale value began at 0.05 (95% UI: 0.05–0.06) among 15-year-olds, peaked among 37-year-olds at 0.11 (95% UI: 0.10–0.11), and decreased back down to 0.05 (95% UI: 0.04–0.06) at age 64.

Finally, *Figure 7* presents inferred distributional parameters from Distributional Model 4 for both men and women for all three data sets. Based on those plots, the flexible model was justified for most distributional parameters in all three data sets. Were we to continue developing these models, this plot suggests that skewness might only need linear, sex-specific effects with respect to age. Interestingly, the 2016–2017 Haiti DHS and Manicaland estimates exhibit similar patterns across all four parameters, despite the different socio-cultural contexts surrounding partnerships in the two populations. We also note that the DHS does not collect data on adults aged 50 years and older, so our estimates in Haiti from age 50 to age 64 are purely extrapolated.

**Table 5.** ELPD and QQ RMSE values for all five distributional regression models fit to each data set.
The models increase in complexity from Conventional Regression to Distributional Model 4. Bolded ELPD values are more than two standard errors higher than the next best value in the column. Bolded QQ RMSE values are lowest in their column.

| Model | AHRI | Haiti 2016–17 DHS | Manicaland |
|---|---|---|---|
| ELPD | | | |
| Conventional | 52689.2 | 4777.8 | 21011.3 |
| Distributional 1 | 54335.2 | 5140.8 | 23192.5 |
| Distributional 2 | 54794.8 | 5138.7 | 23472.1 |
| Distributional 3 | 55534.2 | 5196.7 | 24313.7 |
| Distributional 4 | **55841.9** | 5207.6 | **24516.1** |
| QQ RMSE | | | |
| Conventional | 1.30 | 1.33 | 2.05 |
| Distributional 1 | 1.15 | 0.98 | 1.89 |
| Distributional 2 | 1.21 | 0.99 | 1.80 |
| Distributional 3 | 0.93 | 0.91 | 1.34 |
| Distributional 4 | **0.66** | **0.84** | **1.04** |

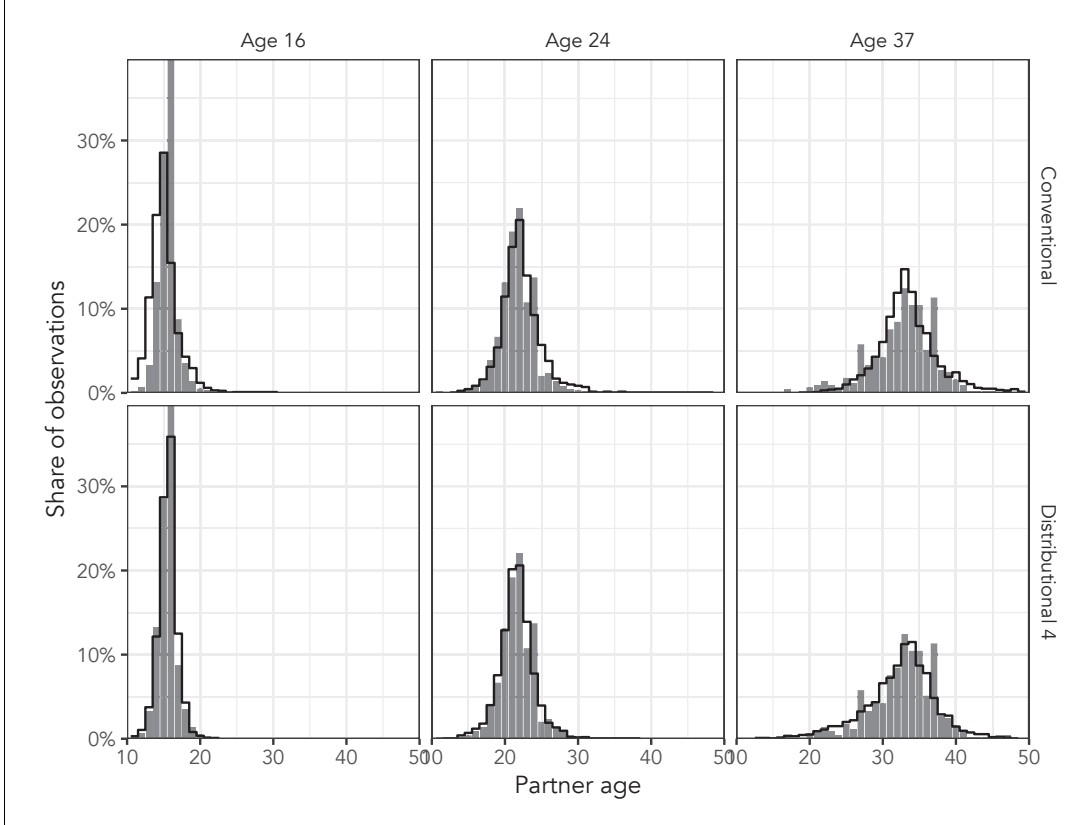

**Figure 5.** Observed partner age distributions (grey bars) and posterior predictive partner age distributions (lines) for conventional regression and the most complex distributional model among men aged 16, 24, and 37 years in the AHRI data set. Posterior predictive distributions come from regression models fit to the entire AHRI data set.

## Discussion

We found that the sinh-arcsinh distribution reproduced observed sexual partner age distributions better than a number of other possible distributional assumptions across age and sex in three distinct data sets. We integrated this finding into a distributional regression framework using existing statistical modelling software. Even the simplest distributional regression in our set of candidate models far outperformed conventional regression, in which all moments except the first are estimated as constants. Our most complex distributional model fit better than all other models in all three data sets, suggesting that modelling these data benefits from the additional complexity.

These results indicate that distributional regression models with sinh-arcsinh distributions can accurately replicate age-/sex-specific sexual partner age distributions. This approach presents a number of advantages over previous methods. First, like Smid et al., it allows a unique distribution for every age-sex combination. As *Figure 3* illustrates, partner age distributions can exhibit substantial, systematic variation across age and sex in any of the first four moments, so we must consider modelling strategies that allow for such variation. Second, distributional regression offers a principled method to propagate uncertainty through this estimation process.

Finally, distributional regression implemented through brms provides access to a deep set of hierarchical modelling tools that could enable estimation in a variety of low-data settings. We evaluated a small set of relatively simple distributional models in this work, but, theoretically, each distributional parameter could have its own, arbitrarily complex hierarchical regression model. Using these tools, one could estimate unique partner age distributions across levels of stratification that are substantively interesting but do not provide sufficient sample size for independent estimation (e.g. study sites or geographic areas).

We have identified several limitations in this approach. First, the amount of data required to produce usefully precise estimates is not tested. Each additional distributional parameter introduces

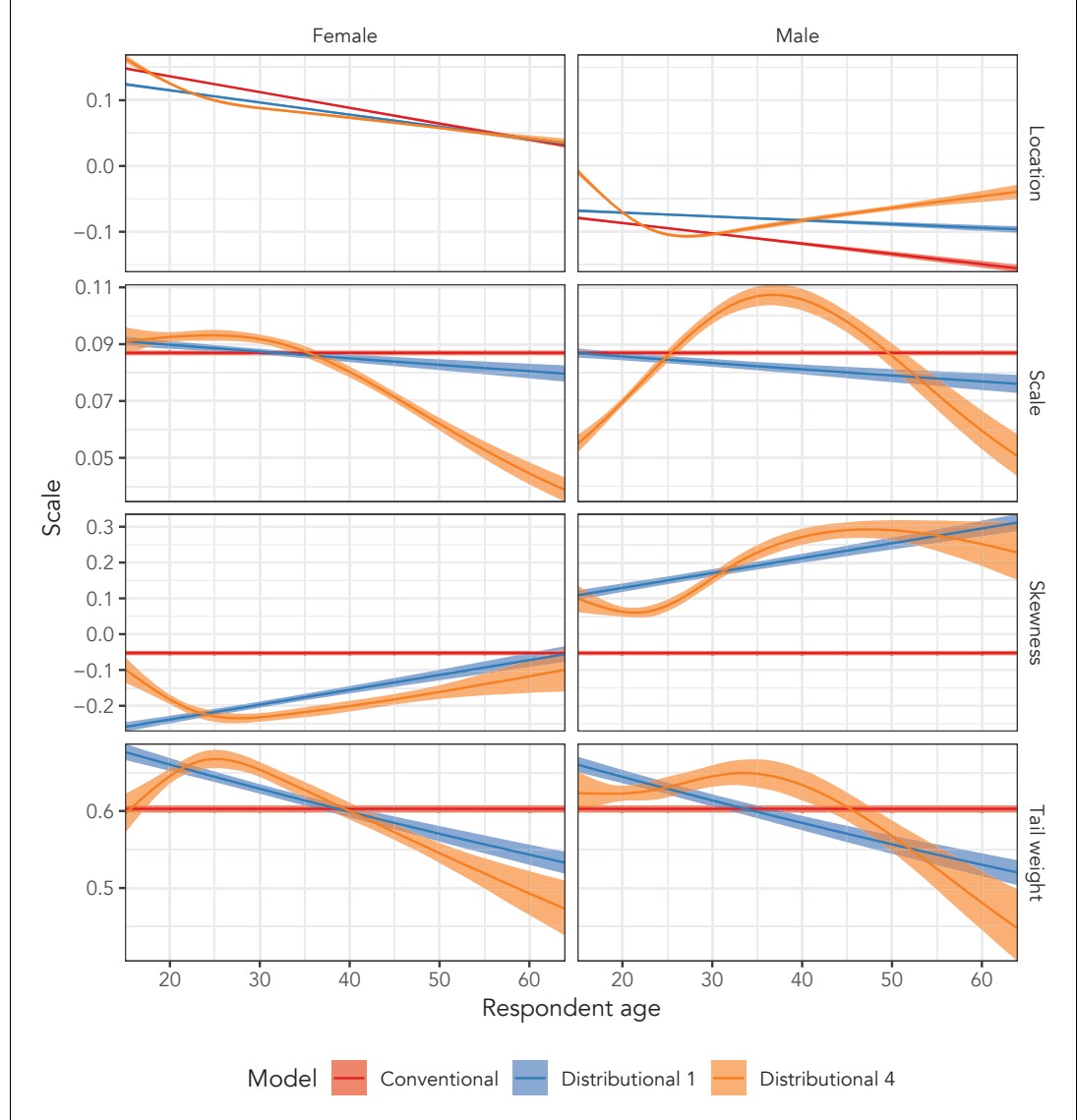

**Figure 6.** Estimated sinh-arcsinh distributional parameters from the conventional regression model, and distributional models 1 and 4 fit to the AHRI data. 'Conventional' assumes no variation across age and sex, 'Distributional 1' allows for independent age and sex effects, and 'Distributional 4' includes sex-specific splines with respect to age.

model parameters, so this method is more complex than conventional regression. The sinh-arcsinh distribution did fail to produce the highest ELPD in our smallest data subset (N = 170), but it was not significantly worse than the best distribution. More importantly, by integrating these data into a distributional modelling framework, we gain the ability to impose structure on these parameters, which could easily offset the cost of any additional model parameters.

Interpreting the inferred model parameters in sinh-arcsinh regression can also be difficult. Although conventional regression estimates the effects of covariates on expected values, the sinh-arcsinh distribution is parametrised in terms of a location parameter. This parameter correlates closely with the central tendency of the distribution, but it is not strictly equal to the mean. We can reparametrise the distribution so that we estimate a mean (and therefore effects of covariates on the expected value), but it is not currently possible in the probabilistic programming software that underlies brms.

Third, our analysis assumed that we were operating at a level of stratification at which partnerships are basically comparable, but any number of factors could lead to fundamentally different

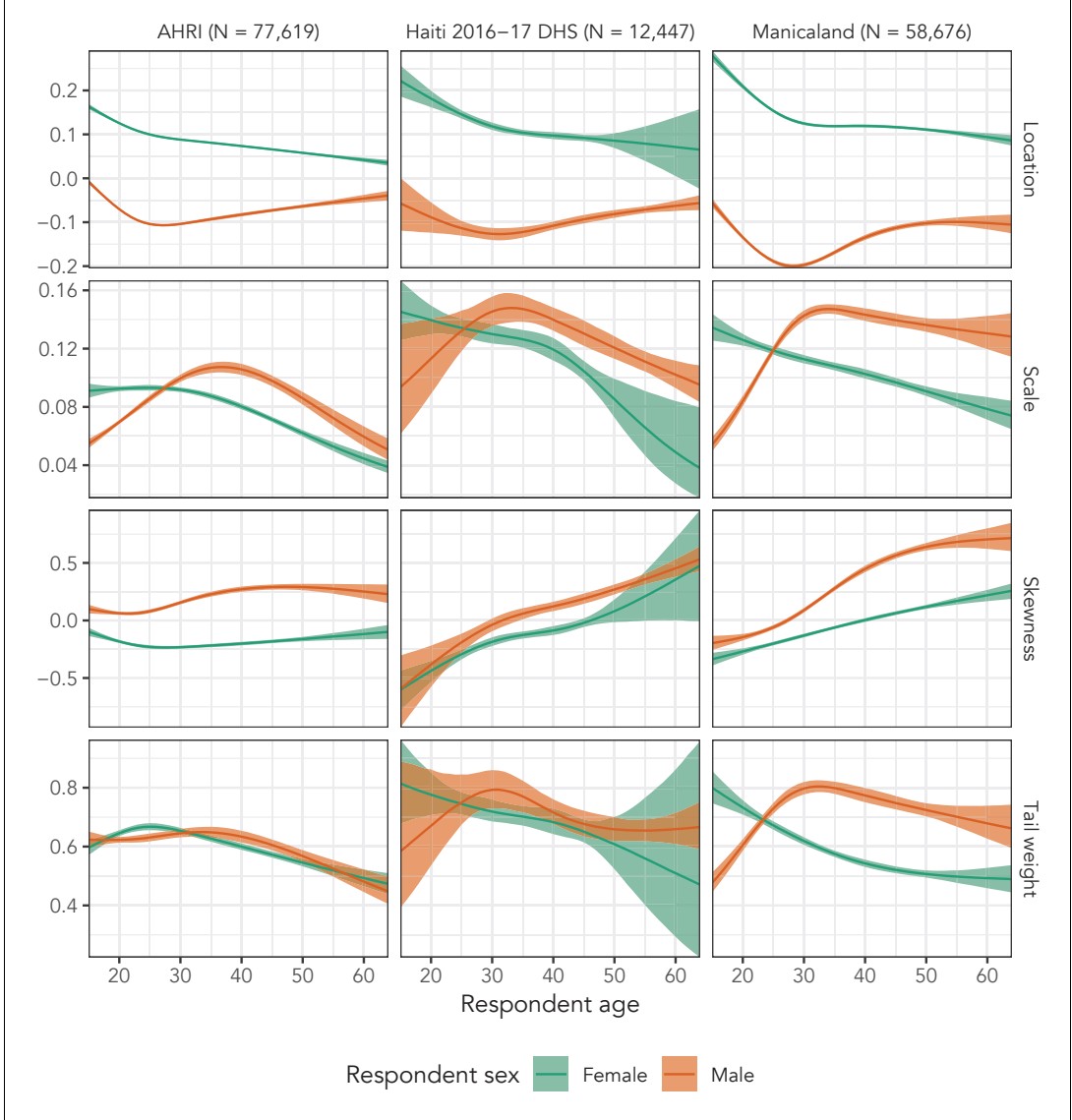

**Figure 7.** Estimated sinh-arcsinh distributional parameters for Distributional Model 4 fit to the three main data sets.

partner age distributions. For example, we did not control for whether the partnership was same-sex or the type of the partnership (married, casual, etc.). That said, our distributional framework would allow us to incorporate data on any of those factors directly into the model.

There are two sources of possible non-independence in our data sets that were not modelled. First, in the two cohort studies, participants are eligible to participate in multiple survey rounds, in which case the same partnership could be reported multiple times in our data set. Second, many individuals report multiple partners in the same survey, and the age of one partner may be correlated ages of other partners, for example due to individual preferences for partnerships. Because we modelled multiple observations of the same individual as conditionally independent, we anticipate that these correlations may artificially increase the precision of our estimates.

Finally, our model does not address any reporting biases in self-reported partnership data. If certain relationship types are perceived as less socially acceptable, respondents might be less likely to report them, resulting in systematic missingness. Our method could still be appropriate to model the age distribution from the data reported about an under-reported partnership type, but it cannot predict whether or not a given partnership exists. However, if under-reporting correlates with

partner age (or age difference), then the empirical distributions will be biased and our method will only smooth and interpolate the biased data.

Despite these limitations, we believe that the strategy we present will work well in future projects that require estimates of partner age distributions. We plan to use these methods to produce age-mixing matrices to inform epidemic models of HIV, but there are a number of additional directions that could be explored. We are specifically interested in leveraging the spatio-temporal structure of the survey data used here. Hierarchical mapping exercises with household survey data are increasingly common in epidemiology, but estimating spatially varying partner age distributions would require an evaluation of how best to model higher order moments over space. We would, for example, need to consider how the variance of partner age distributions varies by urbanicity.

Similarly, population-based studies typically collect far more detailed information on partnerships than we took advantage of here. Relationship type is a key confounder of the association between respondent age and partner age (that we ignored for the purposes of our experiments). We might expect the age distributions of casual partners to vary substantially from those of long-term cohabiting partners. Because we have built our model in an existing regression framework, incorporating new covariates into any of the distributional regression specifications is straightforward. The distribution regression framework with the sinh-arcsinh may also be a useful parametric model for continuous representation of marginal distributions within sexual mixing models or network models, such as the ERGM framework.

We believe that our framework offers a flexible, accurate, and robust method for smoothing and interpolating sexual partner age distributions, but these methods are not specific to partner age distributions. The sinh-arcsinh distribution is relatively easy to implement without incurring high computational cost, so it could be applied in many settings. Even without the distributional regression framework, we have used here, allowing the third and fourth moments of the distribution to vary from the 'default' normal values could be valuable across a variety of applications.

Distributional regression is also underutilised in social science applications. We often work with large surveys that would comfortably support models for higher order parameters. Data requirements will vary by application and model, but, as we have shown here, even a simple distributional model can improve fit and avoid biasing estimates.

## Acknowledgements

JWE, SG, and KAR acknowledge funding support from the Bill and Melinda Gates Foundation (OPP1190661, OPP1164897). JWE and SG acknowledge funding support from the MRC Centre for Global Infectious Disease Analysis (reference MR/R015600/1), jointly funded by the UK Medical Research Council (MRC) and the UK Foreign, Commonwealth and Development Office (FCDO), under the MRC/FCDO Concordat agreement and is also part of the EDCTP2 programme supported by the European Union. JWE also acknowledges funding support from National Institute of Allergy and Infectious Disease of the National Institutes of Health under award number R01AI136664. The content is solely the responsibility of the authors and does not necessarily represent the official views of the National Institutes of Health. SRF acknowledges funding support from the EPSRC (EP/V002910/1). TMW's work is funded by the Imperial College President's PhD Scholarship. We thank all of the people who participated in the three studies that shared their data with us, as well as the survey and data management teams, without whom this work would not be possible.

## Additional information

### Funding

| Funder | Grant reference number | Author |
| --- | --- | --- |
| Bill and Melinda Gates Foundation | OPP1190661 | Jeffrey W Eaton |
| Bill and Melinda Gates Foundation | OPP1164897 | Kathryn A Risher<br>Simon Gregson<br>Jeffrey W Eaton |
| Medical Research Council | MR/R015600/1 | Simon Gregson |

| | | Jeffrey W Eaton |
|---|---|---|
| National Institute of Allergy and Infectious Diseases | R01AI136664 | Jeffrey W Eaton |
| Engineering and Physical Sciences Research Council | EP/V002910/1 | Seth Flaxman |
| Imperial College London | President's PhD Scholarship | Timothy M Wolock |

The funders had no role in study design, data collection and interpretation, or the decision to submit the work for publication.

### Author contributions
Timothy M Wolock, Conceptualization, Software, Formal analysis, Visualization, Methodology, Writing - original draft, Writing - review and editing; Seth Flaxman, Jeffrey W Eaton, Conceptualization, Supervision, Methodology, Writing - review and editing; Kathryn A Risher, Conceptualization, Data curation, Writing - review and editing; Tawanda Dadirai, Simon Gregson, Data curation, Writing - review and editing

### Author ORCIDs
Timothy M Wolock  https://orcid.org/0000-0001-5898-1014
Seth Flaxman  http://orcid.org/0000-0002-2477-4217
Kathryn A Risher  https://orcid.org/0000-0002-9588-1693
Simon Gregson  https://orcid.org/0000-0003-2707-0714
Jeffrey W Eaton  https://orcid.org/0000-0001-7728-728X

### Ethics
Human subjects: We conducted secondary analysis of previously collected anonymised data in compliance with each data producer's use requirements. Procedures and questionnaires for standard DHS surveys have been reviewed and approved by the ICF International Institutional Review Board (IRB). The Manicaland study was approved by the Medical Research Council of Zimbabwe and the Imperial College Research Ethics Committee. The Africa Centre Demographic Information System PIP surveillance study was approved by Biomedical Research Ethics Committee, University of Kwa-Zulu-Natal, South Africa (BE290/16).

### Decision letter and Author response
Decision letter https://doi.org/10.7554/eLife.68318.sa1
Author response https://doi.org/10.7554/eLife.68318.sa2

## Additional files
### Supplementary files
• Transparent reporting form

### Data availability
Data from the Demographic and Health Surveys are available from the DHS Program website (https://dhsprogram.com/data/available-datasets.cfm). Data from the Africa Centre Demographic Information System are available on request from the AHRI website (https://data.ahri.org/index.php/home). Data from the Manicaland study were used with permission from the study investigators (http://www.manicalandhivproject.org/manicaland-data.html).

The following previously published datasets were used:

| Author(s) | Year | Dataset title | Dataset URL | Database and Identifier |
|---|---|---|---|---|
| Gareta D, Dube S, Herbst K | 2020 | AHRI.PIP.Men's General Health.All. Release 2020-07 | https://doi.org/10.23664/AHRI.PIP.RD04-99.MGH. | AHRI Data Repository, 10.23664/AHRI.PIP. |

| | | | | ALL.202007 | RD04-99.MGH.ALL.20 2007 |
|---|---|---|---|---|---|
| Gareta D, Dube S, Herbst K | | 2020 | AHRI.PIP.Women's General Health. All.Release 2020-07 | https://doi.org/10.23664/AHRI.PIP.RD03-99.WGH.ALL.202007 | AHRI Data Repository, 10.23664/AHRI.PIP.RD03-99.WGH.ALL.20 2007 |

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

## Appendix 1

### Age heaping

Respondents in each of these data sets are disproportionately likely to report that their partners' ages are multiples of five or multiples of five away from their own age, leading to distinct 'spikes' in the empirical partner age (or age difference) distributions at multiples of five. The left panel of *Appendix 1—figure 1* illustrates this phenomenon among women aged 24 years in the AHRI data. These spikes, widely referred to as 'heaping', could bias our results towards certain probability distributions, so we developed a simple deheaping algorithm, applied it to the AHRI data.

To account for the possibility that heaping affected the results, we developed a simple deheaping algorithm and treated the deheaped AHRI data as a fourth data set. Due to the structure of the questionnaire ('how many years older or younger is your partner than you?'), the AHRI partner age data exhibit strong heaping on partner ages that are multiples of five years from the respondent's age. For example, among women aged 24 years, we observe far more partners aged exactly 29 years than expected.

Let $n_{s,a,p}$ be the number of observed partnerships with $s_i = s$, $a_i = a$, and $p_i = p$. Fixing age to be $a$ and sex to be $s$, we can find the expected count at partner age $p$, $\hat{n}_{s,a,p}$ by fitting a Nadaraya-Watson estimator to all ordered pairs $(p, n_{s,a,p})$ such that $p - a$ is *not* a multiple of five. We can then find the positive-valued excess counts at all $p$ such that $p - a$ is a multiple of five: $e_{s,a,p} = \max(n_{s,a,p} - \hat{n}_{s,a,p}, 0)$.

This quantity, $e_{s,a,p}$, is what the Nadaraya-Watson estimator has identified as number of heaped observations. Fixing $p^\star$ to be a partner age such that $(p^\star - a) \bmod 5 \equiv 0$, we assume that all of the excess mass at $p^\star$ will be allocated to the four partner ages on either side of $p^\star$. We find the share of $e_{s,a,p^\star}$ to be allocated to each of $(p^\star - 2, ..., p^\star + 2)$, denoted $b_{s,a,p}$, as

$$b_{s,a,p} = \frac{n_{s,a,p}}{\sum_{i=-2}^{2} n_{s,a,p^\star+i}},$$

substituting in $\hat{n}_{s,a,p^\star}$ for $n_{s,a,p}$ wherever applicable. Finally, we find the number of individuals to be reassigned from $p^\star$ to each $p$ within two years of $y^\star$ as $d_{s,a,p} = b_{s,a,p} \cdot e_{s,a,p^\star}$. Note that each partner age can only 'receive' partnerships from its nearest multiple of five and that each multiple of five can only 'send' partnerships to itself and the four partner ages on either side of it. For each $y$ within two years of $y^\star$, we randomly select $\lfloor d_{s,a,p} \rceil$ individuals to move from $p^\star$ to $p$. We apply this method for both sexes and all respondent ages with at least two observations separately.

*Appendix 1—figure 1* illustrates the effect of this process on data among women aged 24 in the AHRI data. This method is quite simple, but it seems to work reasonably well on the AHRI data. Regardless, we do not need a perfect deheaping algorithm for this application; we just need one that will give us a *plausibly* deheaped version of the AHRI data. If the results differ drastically between the heaped and deheaped data sets (i.e. if one probability distribution works perfectly only on the deheaped data), then we will know that our results are sensitive to irregularities in the data.

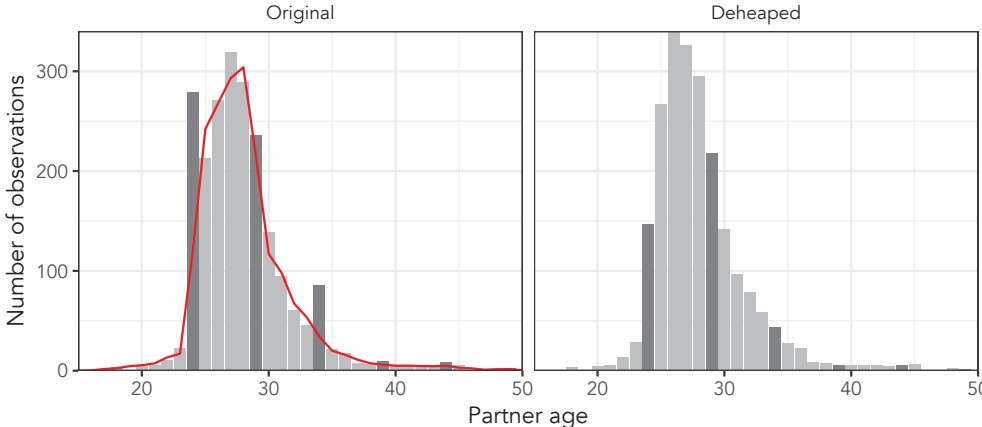

**Appendix 1—figure 1.** Illustration of the effect of the deheaping algorithm on women aged exactly

*Appendix 1—figure 1 continued on next page*

24 years in the AHRI data. Dark grey bars correspond to ages identified as potentially heaped (multiples of five away from 24). The red line is the expected count of observations estimated by excluding any potentially heaped ages.

## Results

*Appendix 1—figure 2* shows the presence of age heaping among women in the AHRI data, as well as the effects of our deheaping algorithm. Visible diagonal lines indicate that women were disproportionately likely to report that the difference between their partner's age and their own age was a multiple of five. Heaping to partner ages (not partner age differences) would manifest as horizontal lines. As we can see in the right panel, the deheaping procedure resolves the majority of the heaping. We cannot validate the algorithm, but for the purposes of this experiment, simply producing plausibly deheaped age distributions should be sufficient.

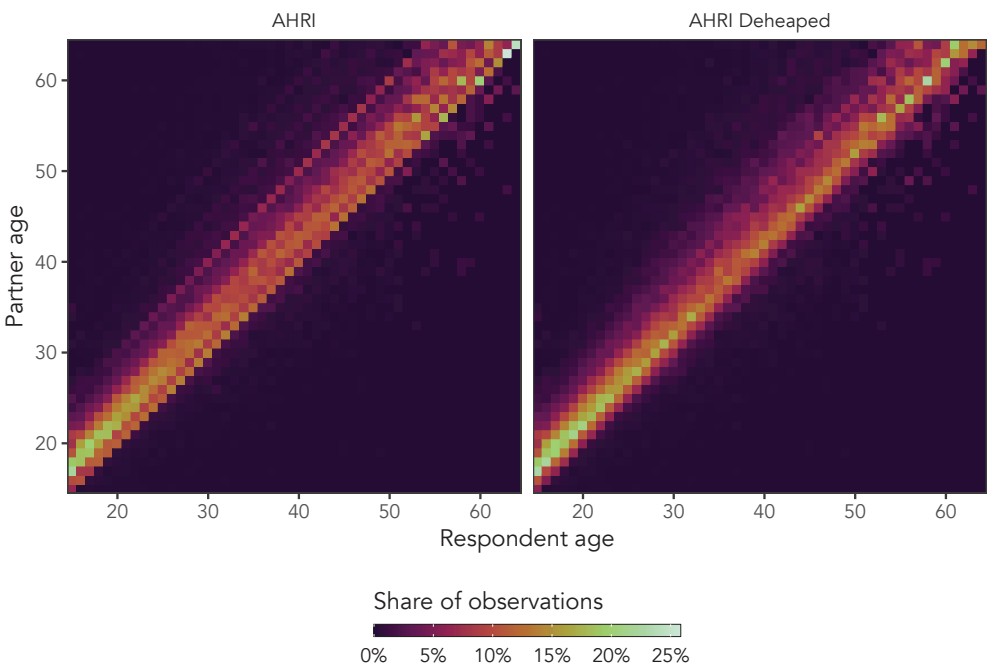

**Appendix 1—figure 2.** Observed sexual partner age distributions among women in the AHRI data. The left panel is original data, and the right panel is the same data set after deheaping age differences from multiples of five.

*Appendix 1—table 1* provides ELPD and QQ RMSE values for all five regression models fit to the deheaped AHRI data. As with the heaped AHRI data, the most complex distributional model had the highest ELPD (58504.0). From these results, we conclude that the presence of heaping in the three main data sets is unlikely to have substantially altered the results of this analysis.

**Appendix 1—table 1.** ELPD and QQ RMSE values for all five models fit to deheaped AHRI data The models increase in complexity from Conventional Regression to Distributional Model 4.
Bolded ELPD values are more than two standard errors higher than the next best value in the column.
Bolded QQ RMSE values are lowest in their column.

| Model | AHRI deheaped |
| --- | --- |
| ELPD | |
| Conventional | 55296.2 |

*Appendix 1—table 1 continued*

| Model | AHRI deheaped |
|---|---|
| Distributional 1 | 57097.4 |
| Distributional 2 | 57503.7 |
| Distributional 3 | 58219.2 |
| **Distributional 4** | **58504.0** |
| QQ RMSE | |
| Conventional | 1.26 |
| Distributional 1 | 1.06 |
| Distributional 2 | 1.14 |
| Distributional 3 | 0.92 |
| **Distributional 4** | **0.62** |

## Model specification details

We modelled the log-ratio dependent variable using the four-parameter sinh-arcsinh distribution:

$$
\begin{aligned}
y_i &\sim \sinh(\mu_i, \sigma_i, \epsilon_i, \delta_i) \\
\mu_i &= \beta^\mu \mathbf{X}_i^\mu \\
\log \sigma_i^\star &= \beta^\sigma \mathbf{X}_i^\sigma \\
\epsilon_i &= \beta^\epsilon \mathbf{X}_i^\epsilon \\
\log \delta_i &= \beta^\delta \mathbf{X}_i^\delta \\
\sigma_i &= \sigma_i^\star \delta_i,
\end{aligned}
$$

where $\beta^\mu$, $\beta^\sigma$, $\beta^\epsilon$, and $\beta^\delta$ are free parameters. We placed essentially arbitrary shrinkage priors on all coefficients:

$$
\beta^\mu, \beta^\sigma, \beta^\epsilon, \beta^\delta \sim \mathrm{N}(0,5).
$$

First, we fit a conventional regression, in which only the location parameter, μ, is a function of data. Specifically, we allowed for linear sex and age effects and a linear interaction between respondent sex and age ($s_i$ and $a_i$, respectively) in the model of μ:

$$
\begin{aligned}
\mathbf{X}_i^\mu &= (1, s_i, a_i, s_i \cdot a_i) \\
\mathbf{X}_i^\sigma, \mathbf{X}_i^\epsilon, \mathbf{X}_i^\delta &= (1).
\end{aligned}
$$

In the second model, we allowed the three higher order distributional parameters to vary by age and sex:

$$
\begin{aligned}
\mathbf{X}_i^\mu &= (1, s_i, a_i, s_i \cdot a_i) \\
\mathbf{X}_i^\sigma, \mathbf{X}_i^\epsilon, \mathbf{X}_i^\delta &= (1, s_i, a_i).
\end{aligned}
$$

In the third model, all four distributional parameters had age, sex, and age-sex interaction effects:

$$
\mathbf{X}_i^\mu, \mathbf{X}_i^\sigma, \mathbf{X}_i^\epsilon, \mathbf{X}_i^\delta = (1, s_i, a_i, s_i \cdot a_i)
$$

To allow for the possibility of non-linear variation with respect to age in the fourth model, we modelled the location parameter using sex-specific natural splines on age:

$$
\begin{aligned}
\mathbf{X}_i^\mu &= (1, s_i, \phi_1(a_i), ..., \phi_K(a_i), s_i \cdot \phi_1(a_i), ..., s_i \cdot \phi_K(a_i)) \\
\mathbf{X}_i^\sigma, \mathbf{X}_i^\epsilon, \mathbf{X}_i^\delta &= (1, s_i, a_i, s_i \cdot a_i),
\end{aligned}
$$

where $K$ is the number of columns in the spline design matrix. By including a second set of basis function values that are multiplied by $s_i$, we are estimating an additional, female-specific trend with respect to age.

Finally, we fit a fifth model, in which all four distributional parameters were modelled as sex-specific splines with respect to age:

$$\mathbf{X}_i^\mu, \mathbf{X}_i^\sigma, \mathbf{X}_i^\epsilon, \mathbf{X}_i^\delta \;=\; (1, s_i, \phi_1(a_i), ..., \phi_K(a_i), s_i \cdot \phi_1(a_i), ..., s_i \cdot \phi_K(a_i)).$$

## Full Results

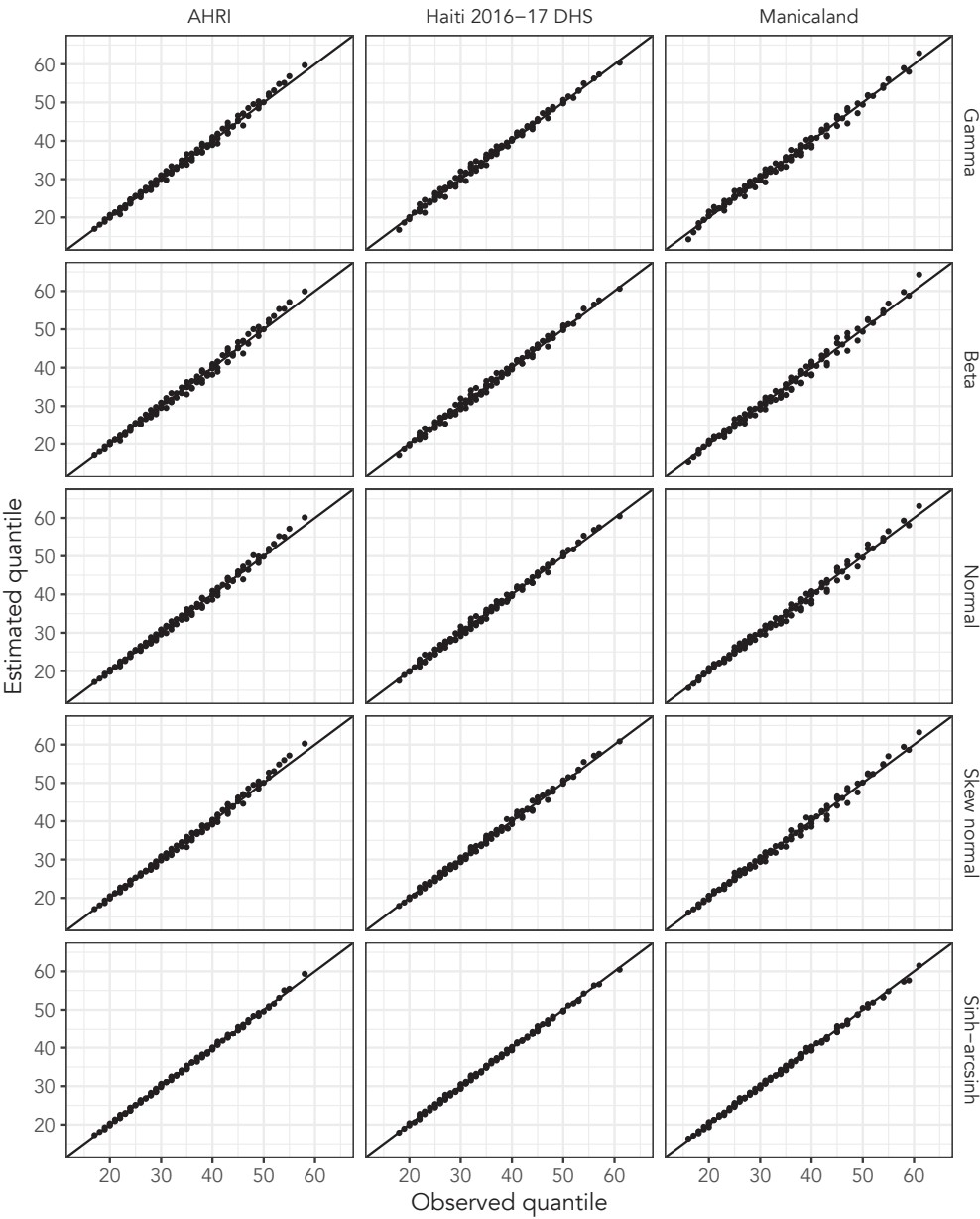

**Appendix 1—figure 3.** Overlaid quantile-quantile (QQ) plots for each probability distribution's best fit to data in all three main data sets. Presented quantiles range from 10th to 90th in increments of 10. Lines closer to the line of equality indicate better fit to empirical quantiles.

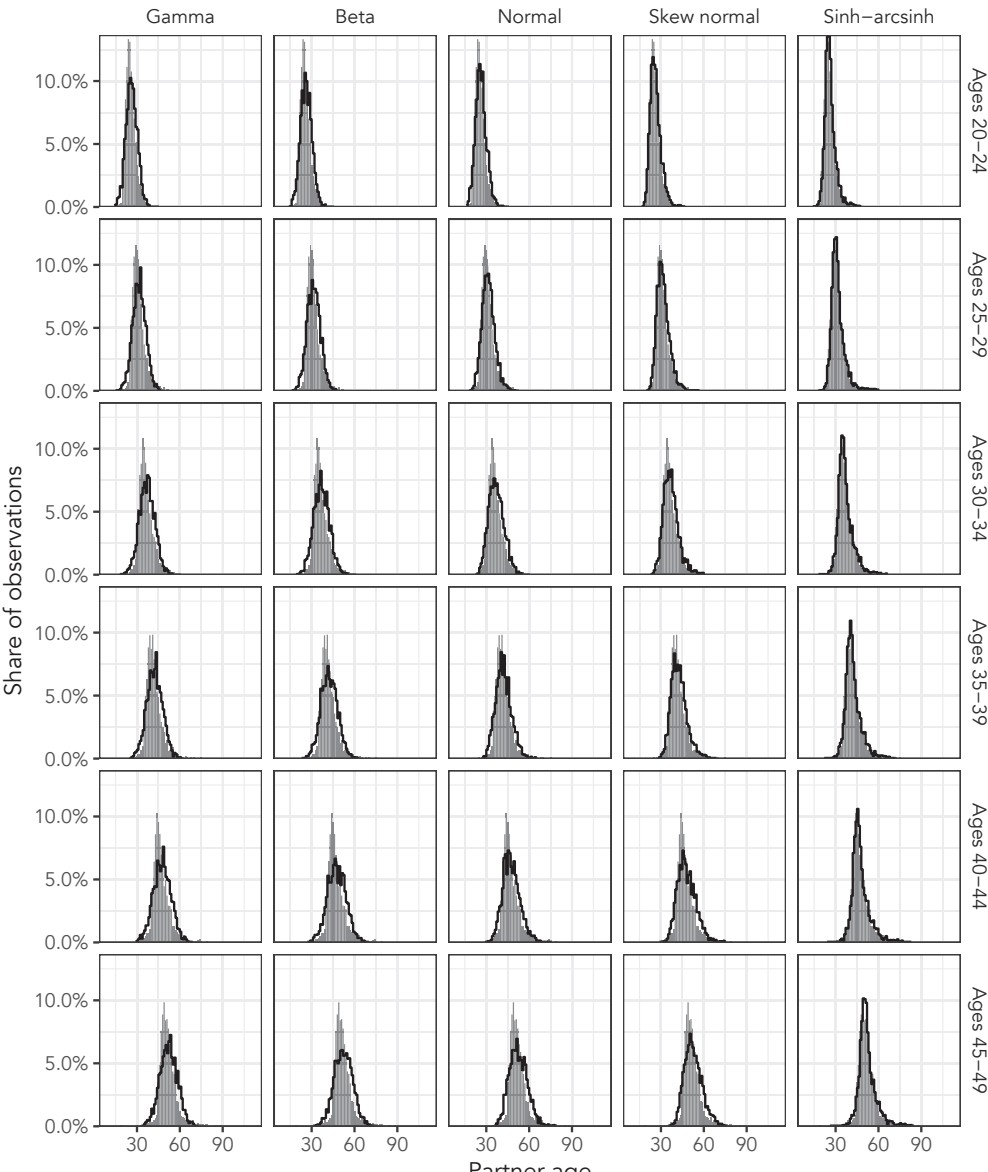

**Appendix 1—figure 4.** Observed partner age distributions (grey bars) and posterior predictive partner age distributions (lines) for each probability distribution among women in the AHRI data set. Here, we plot the posterior predicitve distribution associated with each distribution's highest-ELPD dependent variable.

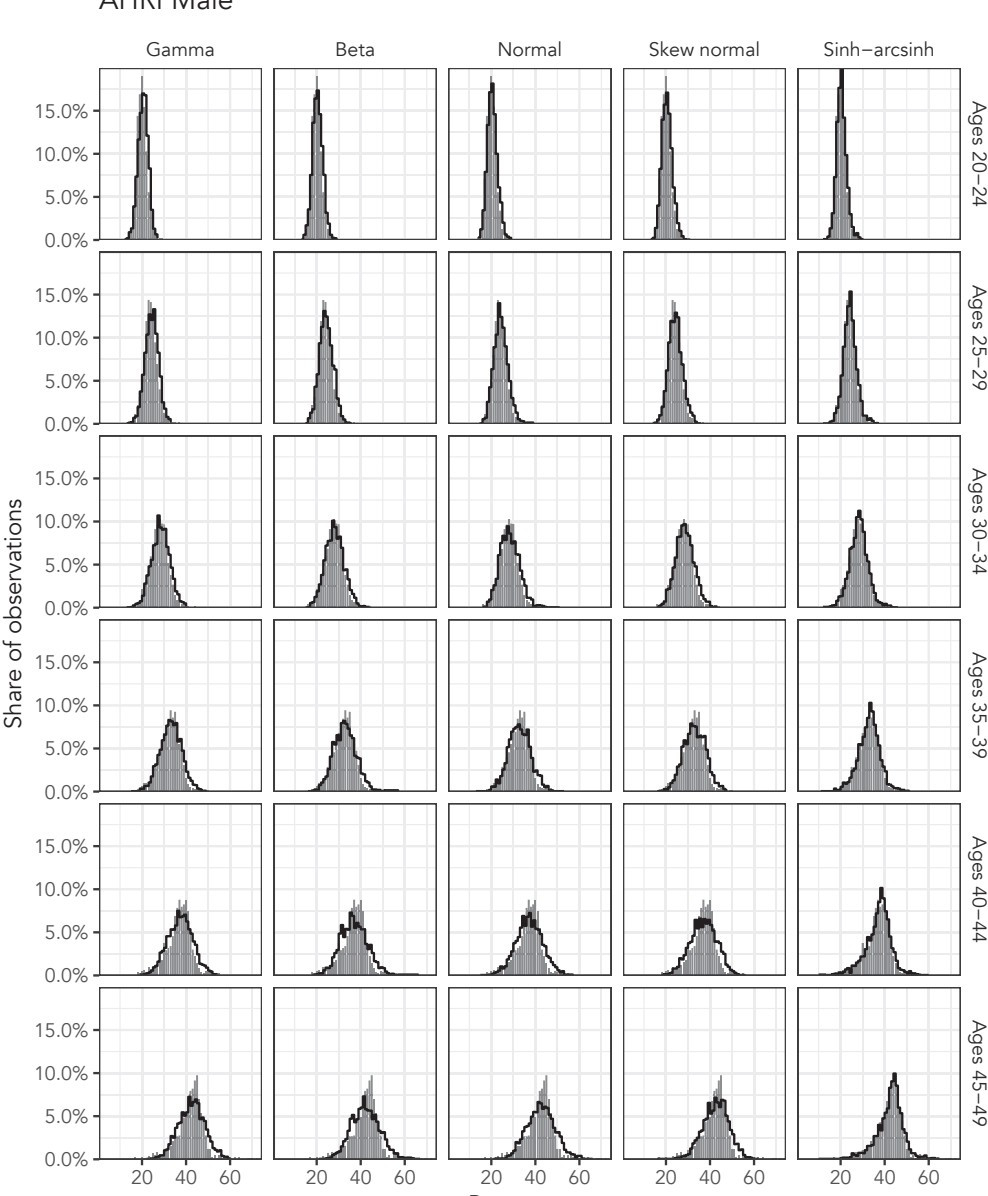

**Appendix 1—figure 5.** Observed partner age distributions (grey bars) and posterior predictive partner age distributions (lines) for each probability distribution among men in the AHRI data set. Here, we plot the posterior predicitve distribution associated with each distribution's highest-ELPD dependent variable.

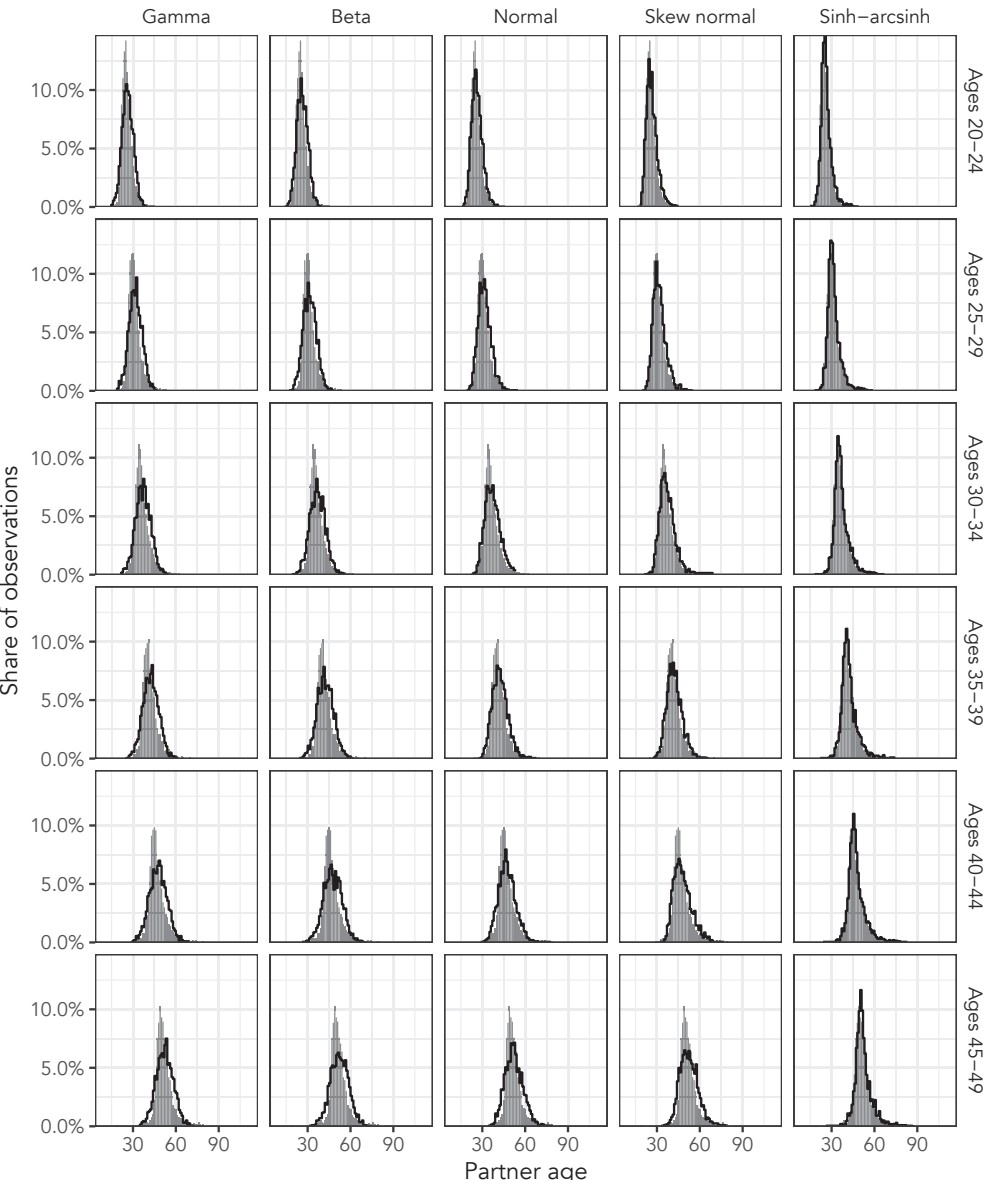

**Appendix 1—figure 6.** Observed partner age distributions (grey bars) and posterior predictive partner age distributions (lines) for each probability distribution among women in the AHRI Deheaped data set. Here, we plot the posterior predicitve distribution associated with each distribution's highest-ELPD dependent variable.

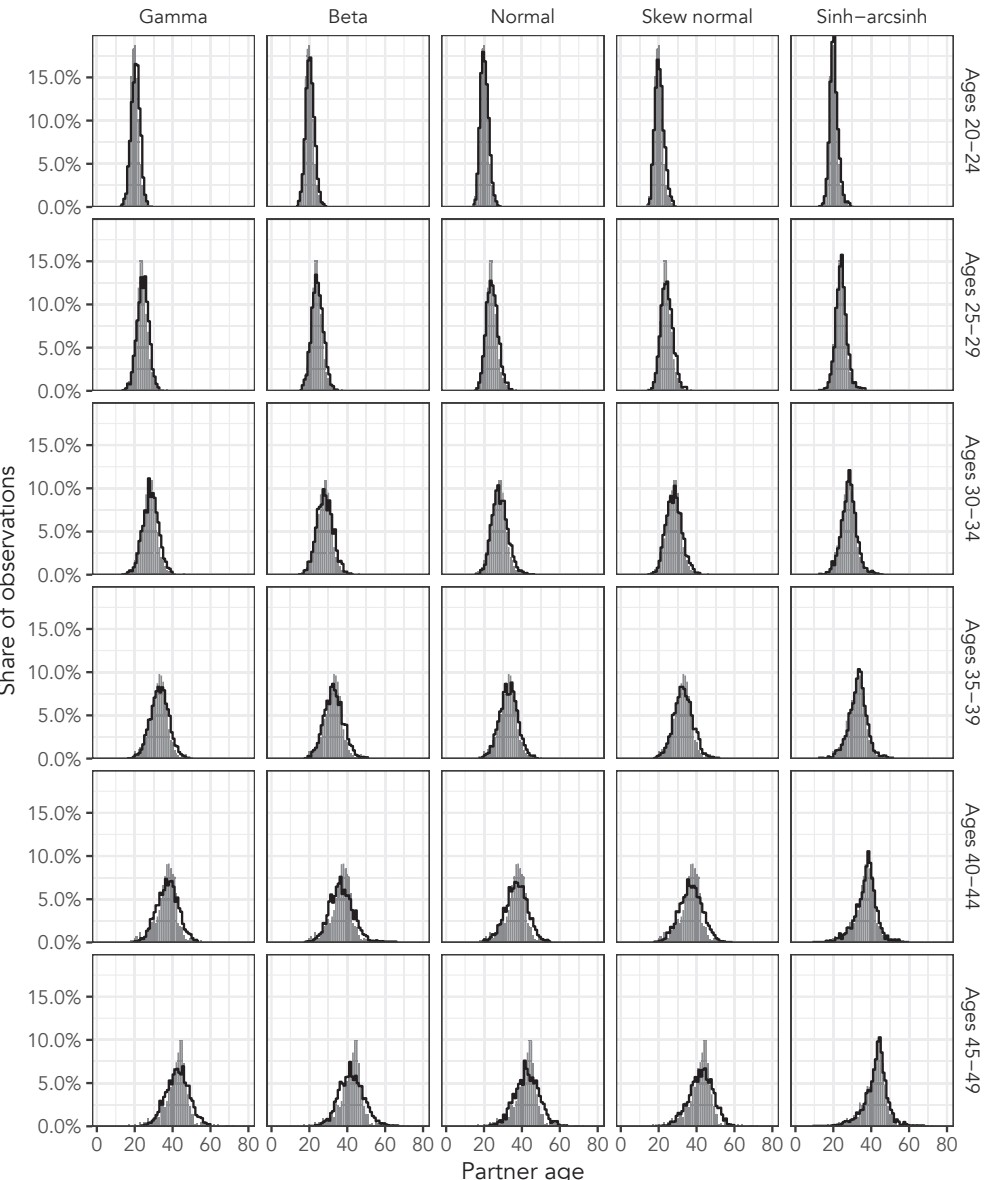

**Appendix 1—figure 7.** Observed partner age distributions (grey bars) and posterior predictive partner age distributions (lines) for each probability distribution among men in the AHRI Deheaped data set. Here, we plot the posterior predicitve distribution associated with each distribution's highest-ELPD dependent variable.

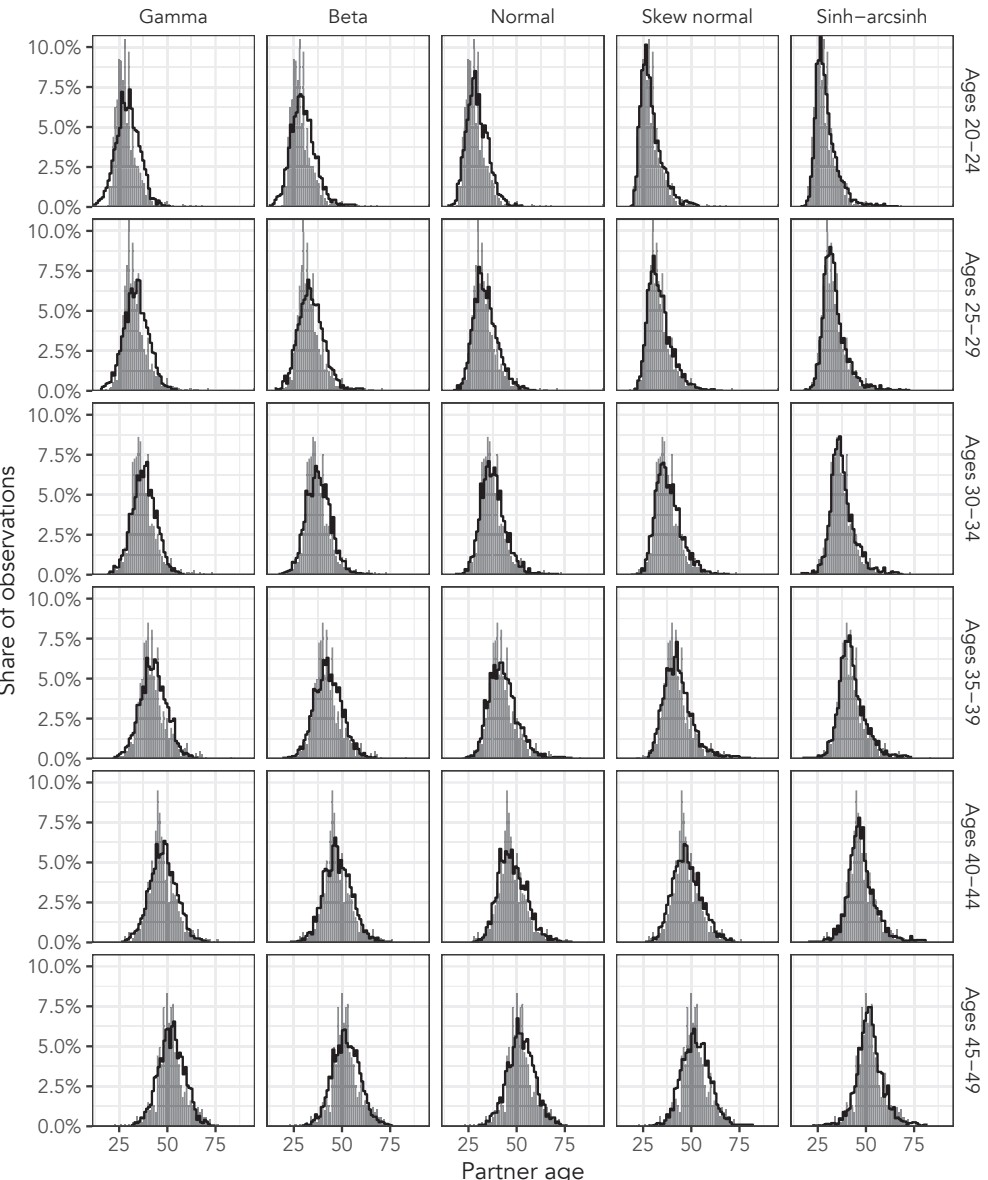

**Appendix 1—figure 8.** Observed partner age distributions (grey bars) and posterior predictive partner age distributions (lines) for each probability distribution among women in the Haiti 2016–17 DHS data set. Here, we plot the posterior predicitve distribution associated with each distribution's highest-ELPD dependent variable.

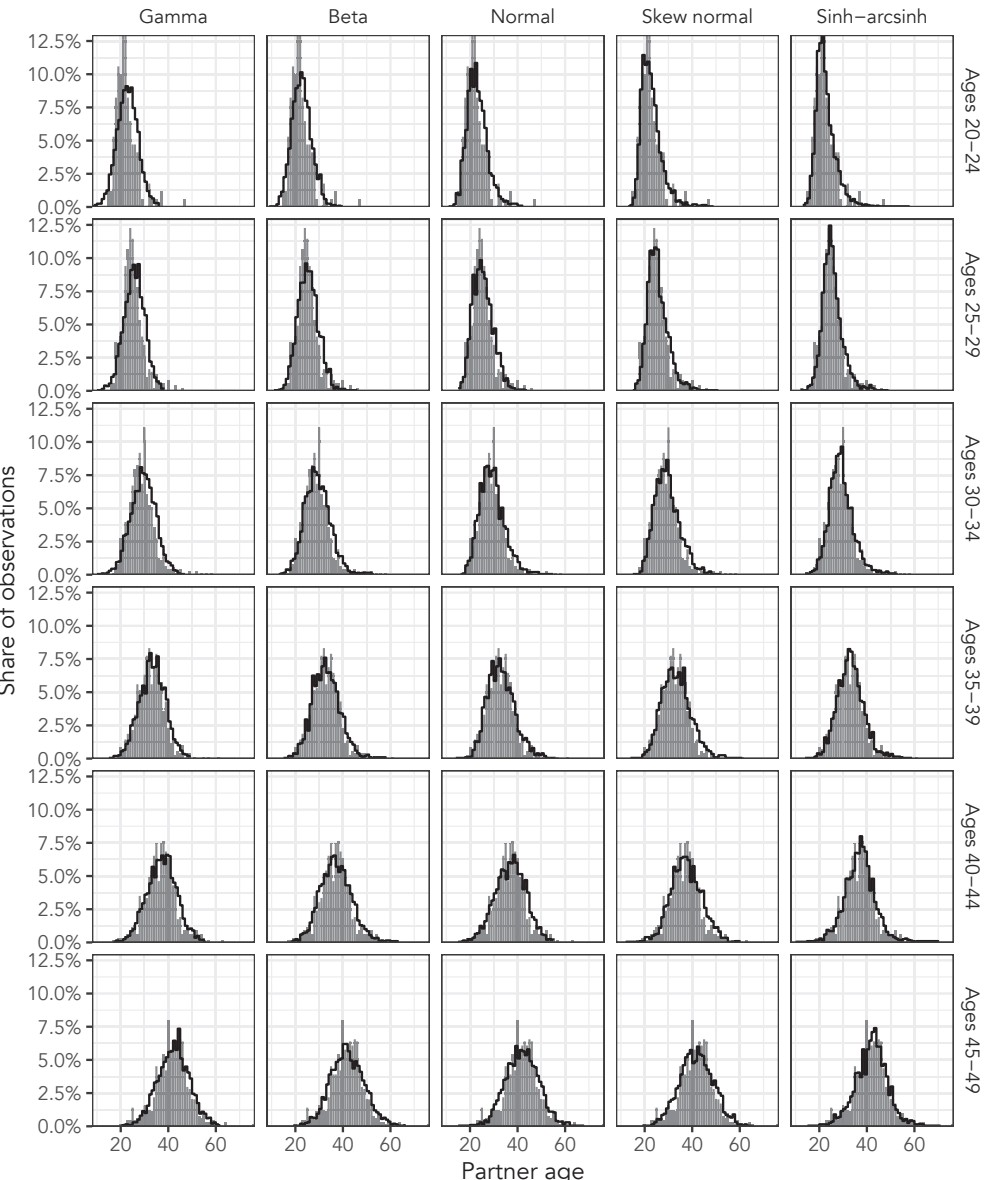

**Appendix 1—figure 9.** Observed partner age distributions (grey bars) and posterior predictive part-
ner age distributions (lines) for each probability distribution among men in the Haiti 2016–17 DHS
data set. Here, we plot the posterior predicitve distribution associated with each distribution's
highest-ELPD dependent variable.

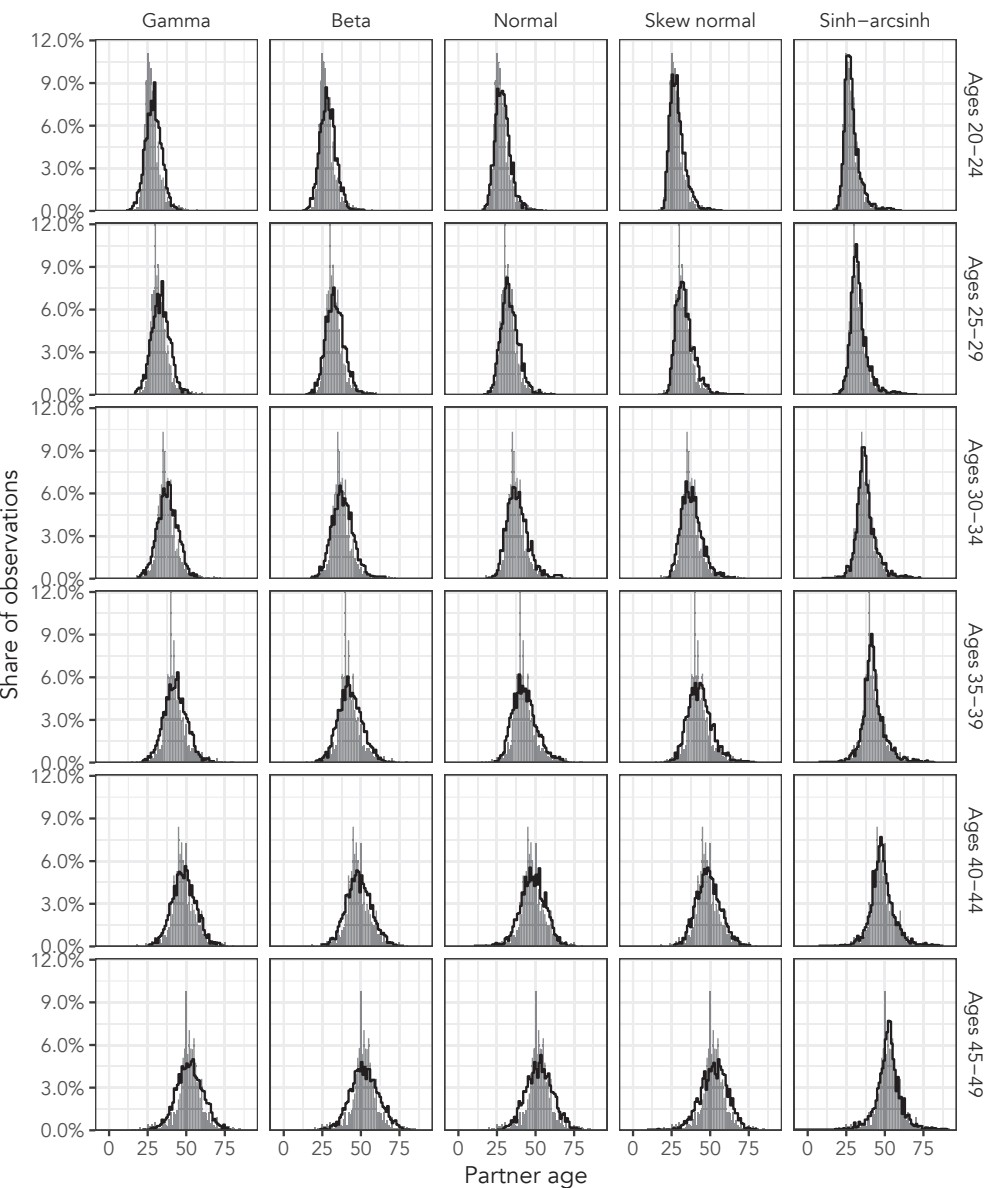

**Appendix 1—figure 10.** Observed partner age distributions (grey bars) and posterior predictive partner age distributions (lines) for each probability distribution among women in the Manicaland data set. Here, we plot the posterior predicitve distribution associated with each distribution's highest-ELPD dependent variable.

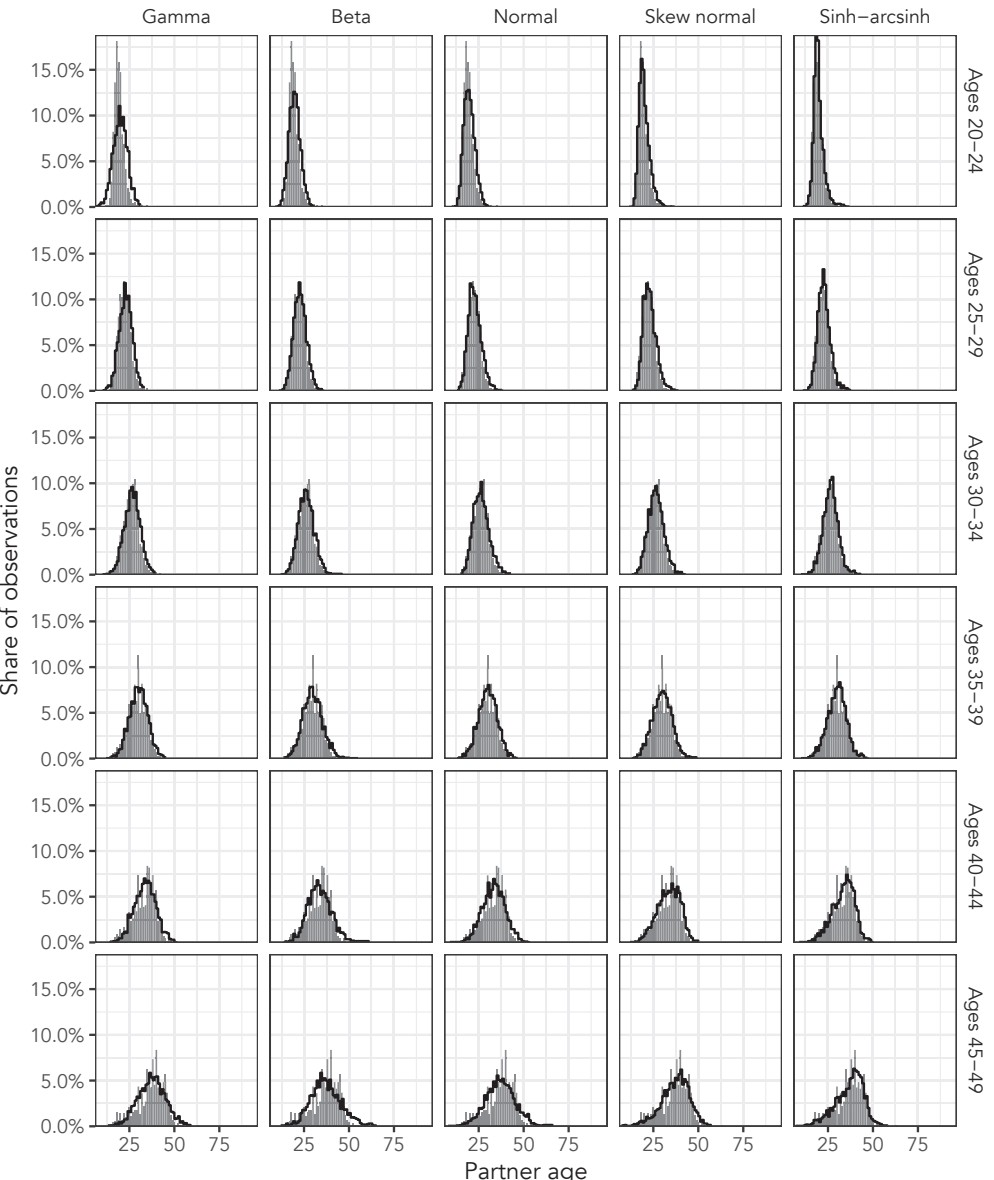

**Appendix 1—figure 11.** Observed partner age distributions (grey bars) and posterior predictive partner age distributions (lines) for each probability distribution among men in the Manicaland data set. Here, we plot the posterior predicitve distribution associated with each distribution's highest-ELPD dependent variable.

**Appendix 1—table 2.** Full ELPD and QQ RMSE table for women in the AHRI data set. Higher ELPD values and lower QQ RMSE values are better.

| Rank | Model | ELPD | ELPD Diff | SE of Diff | QQ RMSE |
|------|-------|------|-----------|------------|---------|
| AHRI Female 20-24 | | | | | |
| 1 | Sinh-arcsinh | −31750.94 | 0.00 | 0.00 | 0.32 |
| 2 | Skew normal | −32056.39 | −305.46 | 48.63 | 0.47 |
| 3 | Normal | −32414.54 | −663.61 | 60.54 | 0.62 |

*Continued on next page*

*Appendix 1—table 2 continued*

| Rank | Model | ELPD | ELPD Diff | SE of Diff | QQ RMSE |
|---|---|---|---|---|---|
| 4 | Beta | −32953.92 | −1202.98 | 112.08 | 0.77 |
| 5 | Gamma | −33461.85 | −1710.92 | 148.15 | 0.80 |
| AHRI Female 25-29 | | | | | |
| 1 | Sinh-arcsinh | −24647.65 | 0.00 | 0.00 | 0.28 |
| 2 | Skew normal | −24906.22 | −258.57 | 43.27 | 0.52 |
| 3 | Normal | −25238.71 | −591.06 | 54.82 | 0.68 |
| 4 | Beta | −25701.13 | −1053.48 | 114.84 | 0.89 |
| 5 | Gamma | −25995.81 | −1348.16 | 132.15 | 0.90 |
| AHRI Female 30-34 | | | | | |
| 1 | Sinh-arcsinh | −19831.53 | 0.00 | 0.00 | 0.44 |
| 2 | Skew normal | −20200.44 | −368.91 | 69.40 | 0.51 |
| 3 | Normal | −20314.79 | −483.26 | 52.24 | 0.80 |
| 4 | Beta | −20575.61 | −744.08 | 67.46 | 0.93 |
| 5 | Gamma | −20708.35 | −876.82 | 73.89 | 0.91 |
| AHRI Female 35-39 | | | | | |
| 1 | Sinh-arcsinh | −15469.18 | 0.00 | 0.00 | 0.31 |
| 2 | Skew normal | −15749.79 | −280.61 | 53.04 | 0.77 |
| 3 | Normal | −15834.32 | −365.14 | 41.23 | 0.80 |
| 4 | Beta | −16026.51 | −557.33 | 53.99 | 1.18 |
| 5 | Gamma | −16087.40 | −618.22 | 57.06 | 1.04 |
| AHRI Female 40-44 | | | | | |
| 1 | Sinh-arcsinh | −12556.61 | 0.00 | 0.00 | 0.45 |
| 2 | Skew normal | −12876.71 | −320.10 | 45.85 | 1.27 |
| 3 | Normal | −12935.34 | −378.73 | 52.38 | 0.92 |
| 4 | Beta | −13137.69 | −581.08 | 69.18 | 1.38 |
| 5 | Gamma | −13150.66 | −594.05 | 62.73 | 1.19 |
| AHRI Female 45-49 | | | | | |
| 1 | Sinh-arcsinh | −10059.21 | 0.00 | 0.00 | 0.59 |
| 2 | Skew normal | −10391.95 | −332.74 | 42.75 | 1.36 |
| 3 | Normal | −10433.64 | −374.43 | 48.91 | 1.53 |
| 4 | Gamma | −10527.00 | −467.79 | 50.72 | 1.35 |
| 5 | Beta | −10545.33 | −486.12 | 56.02 | 1.58 |

**Appendix 1—table 3.** Full ELPD and QQ RMSE table for men in the AHRI data set. Higher ELPD values and lower QQ RMSE values are better.

| Rank | Model | ELPD | ELPD Diff | SE of Diff | QQ RMSE |
|---|---|---|---|---|---|
| AHRI Male 20-24 | | | | | |
| 1 | Sinh-arcsinh | −20428.11 | 0.00 | 0.00 | 0.23 |
| 2 | Skew normal | −20499.86 | −71.75 | 17.12 | 0.25 |
| 3 | Normal | −20503.89 | −75.79 | 16.85 | 0.22 |
| 4 | Beta | −20545.59 | −117.49 | 23.21 | 0.22 |
| 5 | Gamma | −20700.24 | −272.13 | 43.53 | 0.29 |

*Continued on next page*

*Appendix 1—table 3 continued*

| Rank | Model | ELPD | ELPD Diff | SE of Diff | QQ RMSE |
|---|---|---|---|---|---|
| AHRI Male 25-29 | | | | | |
| 1 | Sinh-arcsinh | −12664.21 | 0.00 | 0.00 | 0.26 |
| 2 | Skew normal | −12727.03 | −62.82 | 17.86 | 0.28 |
| 3 | Beta | −12739.03 | −74.82 | 18.65 | 0.31 |
| 4 | Normal | −12753.25 | −89.04 | 19.35 | 0.29 |
| 5 | Gamma | −12788.26 | −124.05 | 35.07 | 0.38 |
| AHRI Male 30-34 | | | | | |
| 1 | Sinh-arcsinh | −9301.03 | 0.00 | 0.00 | 0.29 |
| 2 | Skew normal | −9357.18 | −56.15 | 14.08 | 0.43 |
| 3 | Beta | −9371.86 | −70.83 | 16.48 | 0.37 |
| 4 | Normal | −9385.63 | −84.60 | 14.67 | 0.46 |
| 5 | Gamma | −9419.34 | −118.31 | 35.11 | 0.27 |
| AHRI Male 35-39 | | | | | |
| 1 | Sinh-arcsinh | −6746.89 | 0.00 | 0.00 | 0.30 |
| 2 | Skew normal | −6812.77 | −65.88 | 17.73 | 0.64 |
| 3 | Normal | −6817.86 | −70.97 | 23.24 | 0.70 |
| 4 | Beta | −6830.95 | −84.06 | 17.95 | 0.71 |
| 5 | Gamma | −6832.47 | −85.58 | 32.03 | 0.44 |
| AHRI Male 40-44 | | | | | |
| 1 | Sinh-arcsinh | −4610.95 | 0.00 | 0.00 | 0.35 |
| 2 | Skew normal | −4711.78 | −100.84 | 18.66 | 0.92 |
| 3 | Normal | −4713.78 | −102.83 | 18.82 | 0.78 |
| 4 | Gamma | −4718.28 | −107.33 | 24.83 | 0.63 |
| 5 | Beta | −4742.70 | −131.75 | 17.28 | 1.07 |
| AHRI Male 45-49 | | | | | |
| 1 | Sinh-arcsinh | −3683.47 | 0.00 | 0.00 | 0.34 |
| 2 | Skew normal | −3770.59 | −87.12 | 16.56 | 0.81 |
| 3 | Gamma | −3776.33 | −92.86 | 15.56 | 0.87 |
| 4 | Normal | −3778.84 | −95.37 | 14.50 | 1.17 |
| 5 | Beta | −3805.78 | −122.31 | 17.40 | 1.36 |

**Appendix 1—table 4.** Full ELPD and QQ RMSE table for women in the AHRI Deheaped data set. Higher ELPD values and lower QQ RMSE values are better.

| Rank | Model | ELPD | ELPD Diff | SE of Diff | QQ RMSE |
|---|---|---|---|---|---|
| AHRI Deheaped Female 20-24 | | | | | |
| 1 | Sinh-arcsinh | −31411.24 | 0.00 | 0.00 | 0.26 |
| 2 | Skew normal | −31797.37 | −386.13 | 53.15 | 0.59 |
| 3 | Normal | −32179.29 | −768.05 | 65.50 | 0.56 |
| 4 | Beta | −32737.57 | −1326.32 | 118.47 | 0.76 |
| 5 | Gamma | −33254.17 | −1842.92 | 155.14 | 0.78 |
| AHRI Deheaped Female 25-29 | | | | | |
| 1 | Sinh-arcsinh | −24439.47 | 0.00 | 0.00 | 0.27 |

*Continued on next page*

*Appendix 1—table 4 continued*

| Rank | Model | ELPD | ELPD Diff | SE of Diff | QQ RMSE |
|------|-------|------|-----------|------------|---------|
| 2 | Skew normal | −24768.06 | −328.59 | 46.71 | 0.65 |
| 3 | Normal | −25104.46 | −664.99 | 58.32 | 0.82 |
| 4 | Beta | −25574.33 | −1134.86 | 119.65 | 1.03 |
| 5 | Gamma | −25870.30 | −1430.83 | 137.51 | 1.05 |
| AHRI Deheaped Female 30-34 | | | | | |
| 1 | Sinh-arcsinh | −19680.77 | 0.00 | 0.00 | 0.41 |
| 2 | Skew normal | −20112.70 | −431.94 | 72.95 | 0.55 |
| 3 | Normal | −20228.52 | −547.76 | 56.19 | 0.81 |
| 4 | Beta | −20492.23 | −811.46 | 70.53 | 0.92 |
| 5 | Gamma | −20624.82 | −944.06 | 76.98 | 0.80 |
| AHRI Deheaped Female 35-39 | | | | | |
| 1 | Sinh-arcsinh | −15381.68 | 0.00 | 0.00 | 0.26 |
| 2 | Skew normal | −15703.77 | −322.09 | 55.30 | 0.68 |
| 3 | Normal | −15788.73 | −407.05 | 43.67 | 0.82 |
| 4 | Beta | −15983.57 | −601.90 | 56.31 | 1.13 |
| 5 | Gamma | −16044.22 | −662.54 | 59.17 | 1.04 |
| AHRI Deheaped Female 40-44 | | | | | |
| 1 | Sinh-arcsinh | −12491.91 | 0.00 | 0.00 | 0.25 |
| 2 | Skew normal | −12846.63 | −354.72 | 47.38 | 0.99 |
| 3 | Normal | −12905.04 | −413.12 | 54.14 | 0.89 |
| 4 | Beta | −13109.82 | −617.91 | 70.96 | 1.31 |
| 5 | Gamma | −13121.45 | −629.53 | 64.12 | 1.13 |
| AHRI Deheaped Female 45-49 | | | | | |
| 1 | Sinh-arcsinh | −9981.83 | 0.00 | 0.00 | 0.53 |
| 2 | Skew normal | −10357.85 | −376.01 | 45.08 | 1.43 |
| 3 | Normal | −10401.64 | −419.80 | 51.57 | 1.46 |
| 4 | Gamma | −10493.73 | −511.90 | 52.90 | 1.37 |
| 5 | Beta | −10513.46 | −531.63 | 58.21 | 1.61 |

**Appendix 1—table 5.** Full ELPD and QQ RMSE table for men in the AHRI Deheaped data set. Higher ELPD values and lower QQ RMSE values are better.

| Rank | Model | ELPD | ELPD Diff | SE of Diff | QQ RMSE |
|------|-------|------|-----------|------------|---------|
| AHRI Deheaped Male 20-24 | | | | | |
| 1 | Sinh-arcsinh | −20310.35 | 0.00 | 0.00 | 0.27 |
| 2 | Skew normal | −20429.90 | −119.55 | 27.09 | 0.22 |
| 3 | Normal | −20459.73 | −149.38 | 35.78 | 0.29 |
| 4 | Beta | −20574.15 | −263.80 | 75.13 | 0.22 |
| 5 | Gamma | −20899.52 | −589.17 | 175.99 | 0.27 |
| AHRI Deheaped Male 25-29 | | | | | |
| 1 | Sinh-arcsinh | −12585.54 | 0.00 | 0.00 | 0.28 |
| 2 | Skew normal | −12680.59 | −95.05 | 21.53 | 0.44 |
| 3 | Beta | −12697.00 | −111.46 | 23.31 | 0.37 |

*Continued on next page*

*Appendix 1—table 5 continued*

| Rank | Model | ELPD | ELPD Diff | SE of Diff | QQ RMSE |
|---|---|---|---|---|---|
| 4 | Normal | −12701.76 | −116.23 | 22.96 | 0.41 |
| 5 | Gamma | −12763.81 | −178.27 | 41.24 | 0.39 |
| AHRI Deheaped Male 30-34 | | | | | |
| 1 | Sinh-arcsinh | −9227.26 | 0.00 | 0.00 | 0.37 |
| 2 | Skew normal | −9302.42 | −75.16 | 16.15 | 0.41 |
| 3 | Beta | −9318.24 | −90.97 | 19.07 | 0.39 |
| 4 | Normal | −9327.58 | −100.31 | 16.18 | 0.41 |
| 5 | Gamma | −9372.32 | −145.06 | 38.27 | 0.27 |
| AHRI Deheaped Male 35-39 | | | | | |
| 1 | Sinh-arcsinh | −6694.86 | 0.00 | 0.00 | 0.30 |
| 2 | Skew normal | −6774.11 | −79.26 | 19.32 | 0.61 |
| 3 | Normal | −6780.69 | −85.84 | 25.42 | 0.44 |
| 4 | Beta | −6791.95 | −97.10 | 19.81 | 0.69 |
| 5 | Gamma | −6796.41 | −101.55 | 34.45 | 0.40 |
| AHRI Deheaped Male 40-44 | | | | | |
| 1 | Sinh-arcsinh | −4591.04 | 0.00 | 0.00 | 0.49 |
| 2 | Skew normal | −4700.54 | −109.51 | 19.38 | 1.16 |
| 3 | Normal | −4703.52 | −112.49 | 19.93 | 1.00 |
| 4 | Gamma | −4708.43 | −117.40 | 25.94 | 0.89 |
| 5 | Beta | −4731.41 | −140.37 | 17.84 | 1.30 |
| AHRI Deheaped Male 45-49 | | | | | |
| 1 | Sinh-arcsinh | −3680.18 | 0.00 | 0.00 | 0.30 |
| 2 | Normal | −3796.06 | −115.88 | 19.24 | 1.15 |
| 3 | Skew normal | −3797.14 | −116.95 | 23.48 | 1.02 |
| 4 | Gamma | −3801.02 | −120.83 | 24.51 | 0.98 |
| 5 | Beta | −3817.97 | −137.79 | 19.37 | 1.39 |

**Appendix 1—table 6.** Full ELPD and QQ RMSE table for women in the Haiti 2016–17 DHS data set. Higher ELPD values and lower QQ RMSE values are better.

| Rank | Model | ELPD | ELPD Diff | SE of Diff | QQ RMSE |
|---|---|---|---|---|---|
| Haiti 2016-17 DHS Female 20-24 | | | | | |
| 1 | Sinh-arcsinh | −3259.31 | 0.00 | 0.00 | 0.49 |
| 2 | Skew normal | −3263.46 | −4.15 | 4.95 | 0.53 |
| 3 | Normal | −3338.23 | −78.92 | 19.54 | 0.91 |
| 4 | Beta | −3441.91 | −182.60 | 45.77 | 1.24 |
| 5 | Gamma | −3504.85 | −245.54 | 53.90 | 1.29 |
| Haiti 2016-17 DHS Female 25-29 | | | | | |
| 1 | Sinh-arcsinh | −4447.43 | 0.00 | 0.00 | 0.26 |
| 2 | Skew normal | −4471.22 | −23.78 | 8.41 | 0.57 |
| 3 | Normal | −4527.25 | −79.82 | 18.72 | 0.86 |
| 4 | Beta | −4625.97 | −178.54 | 40.88 | 1.23 |
| 5 | Gamma | −4678.20 | −230.77 | 45.81 | 1.22 |

*Continued on next page*

*Appendix 1—table 6 continued*

| Rank | Model | ELPD | ELPD Diff | SE of Diff | QQ RMSE |
|------|-------|------|-----------|------------|---------|
| Haiti 2016-17 DHS Female 30-34 | | | | | |
| 1 | Sinh-arcsinh | −4720.12 | 0.00 | 0.00 | 0.44 |
| 2 | Skew normal | −4749.57 | −29.45 | 9.06 | 0.68 |
| 3 | Normal | −4763.78 | −43.66 | 10.51 | 0.62 |
| 4 | Beta | −4809.11 | −88.99 | 17.19 | 0.85 |
| 5 | Gamma | −4836.82 | −116.70 | 20.32 | 0.83 |
| Haiti 2016-17 DHS Female 35-39 | | | | | |
| 1 | Sinh-arcsinh | −4490.82 | 0.00 | 0.00 | 0.33 |
| 2 | Skew normal | −4518.58 | −27.75 | 8.14 | 0.57 |
| 3 | Normal | −4526.55 | −35.73 | 8.59 | 0.73 |
| 4 | Beta | −4561.27 | −70.45 | 13.60 | 0.94 |
| 5 | Gamma | −4577.84 | −87.01 | 15.27 | 0.86 |
| Haiti 2016-17 DHS Female 40-44 | | | | | |
| 1 | Sinh-arcsinh | −3601.02 | 0.00 | 0.00 | 0.35 |
| 2 | Skew normal | −3629.45 | −28.43 | 7.51 | 0.83 |
| 3 | Normal | −3633.14 | −32.11 | 7.96 | 0.71 |
| 4 | Beta | −3641.61 | −40.59 | 9.76 | 0.73 |
| 5 | Gamma | −3644.89 | −43.86 | 10.47 | 0.64 |
| Haiti 2016-17 DHS Female 45-49 | | | | | |
| 1 | Sinh-arcsinh | −3106.27 | 0.00 | 0.00 | 0.39 |
| 2 | Skew normal | −3133.10 | −26.82 | 7.68 | 0.88 |
| 3 | Gamma | −3133.61 | −27.33 | 7.50 | 0.68 |
| 4 | Normal | −3134.62 | −28.35 | 7.46 | 0.81 |
| 5 | Beta | −3136.89 | −30.62 | 8.61 | 0.88 |

**Appendix 1—table 7.** Full ELPD and QQ RMSE table for men in the Haiti 2016–17 DHS data set. Higher ELPD values and lower QQ RMSE values are better.

| Rank | Model | ELPD | ELPD Diff | SE of Diff | QQ RMSE |
|------|-------|------|-----------|------------|---------|
| Haiti 2016-17 DHS Male 20-24 | | | | | |
| 1 | Skew normal | −468.98 | 0.00 | 0.00 | 0.43 |
| 2 | Sinh-arcsinh | −469.60 | −0.62 | 1.12 | 0.41 |
| 3 | Normal | −475.31 | −6.33 | 4.28 | 0.67 |
| 4 | Beta | −483.53 | −14.55 | 7.33 | 0.65 |
| 5 | Gamma | −500.53 | −31.55 | 13.35 | 0.94 |
| Haiti 2016-17 DHS Male 25-29 | | | | | |
| 1 | Sinh-arcsinh | −1386.13 | 0.00 | 0.00 | 0.38 |
| 2 | Skew normal | −1390.54 | −4.41 | 3.19 | 0.49 |
| 3 | Normal | −1395.47 | −9.34 | 4.79 | 0.60 |
| 4 | Beta | −1407.46 | −21.32 | 7.31 | 0.62 |
| 5 | Gamma | −1434.18 | −48.04 | 11.75 | 0.79 |
| Haiti 2016-17 DHS Male 30-34 | | | | | |
| 1 | Sinh-arcsinh | −2217.20 | 0.00 | 0.00 | 0.44 |

*Continued on next page*

*Appendix 1—table 7 continued*

| Rank | Model | ELPD | ELPD Diff | SE of Diff | QQ RMSE |
|---|---|---|---|---|---|
| 2 | Skew normal | −2222.10 | −4.89 | 3.42 | 0.69 |
| 3 | Normal | −2223.97 | −6.76 | 4.48 | 0.45 |
| 4 | Beta | −2240.58 | −23.37 | 9.32 | 0.52 |
| 5 | Gamma | −2281.18 | −63.98 | 17.91 | 0.73 |
| Haiti 2016-17 DHS Male 35-39 | | | | | |
| 1 | Sinh-arcsinh | −2185.96 | 0.00 | 0.00 | 0.28 |
| 2 | Skew normal | −2189.87 | −3.91 | 2.67 | 0.69 |
| 3 | Beta | −2191.05 | −5.10 | 3.68 | 0.48 |
| 4 | Normal | −2191.11 | −5.16 | 3.55 | 0.49 |
| 5 | Gamma | −2205.69 | −19.73 | 9.57 | 0.52 |
| Haiti 2016-17 DHS Male 40-44 | | | | | |
| 1 | Sinh-arcsinh | −2051.62 | 0.00 | 0.00 | 0.39 |
| 2 | Skew normal | −2060.16 | −8.54 | 4.21 | 0.72 |
| 3 | Normal | −2060.38 | −8.75 | 4.57 | 0.69 |
| 4 | Beta | −2062.00 | −10.37 | 4.87 | 0.70 |
| 5 | Gamma | −2063.79 | −12.17 | 5.73 | 0.47 |
| Haiti 2016-17 DHS Male 45-49 | | | | | |
| 1 | Sinh-arcsinh | −2138.34 | 0.00 | 0.00 | 0.23 |
| 2 | Normal | −2150.53 | −12.19 | 6.38 | 0.35 |
| 3 | Skew normal | −2151.51 | −13.17 | 5.97 | 0.56 |
| 4 | Gamma | −2152.88 | −14.54 | 8.93 | 0.25 |
| 5 | Beta | −2156.13 | −17.79 | 6.14 | 0.48 |

**Appendix 1—table 8.** Full ELPD and QQ RMSE table for women in the Manicaland data set. Higher ELPD values and lower QQ RMSE values are better.

| Rank | Model | ELPD | ELPD Diff | SE of Diff | QQ RMSE |
|---|---|---|---|---|---|
| Manicaland Female 20-24 | | | | | |
| 1 | Sinh-arcsinh | −16390.77 | 0.00 | 0.00 | 0.31 |
| 2 | Skew normal | −16502.01 | −111.25 | 21.22 | 0.44 |
| 3 | Normal | −16779.93 | −389.16 | 37.05 | 0.67 |
| 4 | Beta | −17111.57 | −720.80 | 62.02 | 0.86 |
| 5 | Gamma | −17387.38 | −996.61 | 76.80 | 1.02 |
| Manicaland Female 25-29 | | | | | |
| 1 | Sinh-arcsinh | −18702.50 | 0.00 | 0.00 | 0.53 |
| 2 | Skew normal | −18923.04 | −220.53 | 25.27 | 0.94 |
| 3 | Normal | −19080.66 | −378.16 | 36.05 | 0.83 |
| 4 | Beta | −19405.80 | −703.30 | 64.97 | 1.05 |
| 5 | Gamma | −19615.53 | −913.03 | 76.38 | 1.09 |
| Manicaland Female 30-34 | | | | | |
| 1 | Sinh-arcsinh | −16523.81 | 0.00 | 0.00 | 0.48 |
| 2 | Skew normal | −16877.96 | −354.15 | 40.36 | 0.87 |
| 3 | Normal | −16886.62 | −362.80 | 36.41 | 0.99 |

*Continued on next page*

*Appendix 1—table 8 continued*

| Rank | Model | ELPD | ELPD Diff | SE of Diff | QQ RMSE |
|------|-------|------|-----------|------------|---------|
| 4 | Beta | −17021.26 | −497.44 | 43.60 | 1.12 |
| 5 | Gamma | −17094.58 | −570.76 | 49.53 | 0.93 |
| Manicaland Female 35-39 | | | | | |
| 1 | Sinh-arcsinh | −14397.76 | 0.00 | 0.00 | 0.48 |
| 2 | Skew normal | −14736.64 | −338.88 | 28.35 | 1.25 |
| 3 | Normal | −14798.55 | −400.79 | 36.87 | 1.39 |
| 4 | Beta | −14824.80 | −427.04 | 33.02 | 1.47 |
| 5 | Gamma | −14835.11 | −437.35 | 34.49 | 1.14 |
| Manicaland Female 40-44 | | | | | |
| 1 | Sinh-arcsinh | −12293.13 | 0.00 | 0.00 | 0.68 |
| 2 | Skew normal | −12488.28 | −195.15 | 21.36 | 1.03 |
| 3 | Gamma | −12500.93 | −207.80 | 22.18 | 1.03 |
| 4 | Normal | −12508.91 | −215.78 | 23.29 | 1.28 |
| 5 | Beta | −12537.14 | −244.01 | 25.41 | 1.22 |
| Manicaland Female 45-49 | | | | | |
| 1 | Sinh-arcsinh | −9183.03 | 0.00 | 0.00 | 0.56 |
| 2 | Skew normal | −9455.87 | −272.83 | 23.57 | 1.68 |
| 3 | Normal | −9477.33 | −294.30 | 23.55 | 1.62 |
| 4 | Gamma | −9497.31 | −314.27 | 25.08 | 1.44 |
| 5 | Beta | −9576.44 | −393.40 | 32.07 | 1.94 |

**Appendix 1—table 9.** Full ELPD and QQ RMSE table for men in the Manicaland data set. Higher ELPD values and lower QQ RMSE values are better.

| Rank | Model | ELPD | ELPD Diff | SE of Diff | QQ RMSE |
|------|-------|------|-----------|------------|---------|
| Manicaland Male 20-24 | | | | | |
| 1 | Sinh-arcsinh | −9770.00 | 0.00 | 0.00 | 0.30 |
| 2 | Skew normal | −9895.82 | −125.83 | 33.35 | 0.40 |
| 3 | Normal | −10139.11 | −369.11 | 79.13 | 0.49 |
| 4 | Beta | −10587.64 | −817.64 | 181.23 | 0.56 |
| 5 | Gamma | −11594.58 | −1824.59 | 388.26 | 1.15 |
| Manicaland Male 25-29 | | | | | |
| 1 | Sinh-arcsinh | −13978.59 | 0.00 | 0.00 | 0.40 |
| 2 | Skew normal | −13990.39 | −11.80 | 8.51 | 0.48 |
| 3 | Normal | −14018.60 | −40.00 | 17.48 | 0.45 |
| 4 | Beta | −14152.35 | −173.76 | 48.77 | 0.40 |
| 5 | Gamma | −14500.47 | −521.87 | 117.58 | 0.55 |
| Manicaland Male 30-34 | | | | | |
| 1 | Sinh-arcsinh | −12949.24 | 0.00 | 0.00 | 0.31 |
| 2 | Skew normal | −13016.44 | −67.21 | 25.01 | 0.37 |
| 3 | Normal | −13037.46 | −88.22 | 16.57 | 0.49 |
| 4 | Beta | −13070.31 | −121.07 | 54.74 | 0.41 |
| 5 | Gamma | −13285.47 | −336.23 | 171.92 | 0.42 |

*Continued on next page*

*Appendix 1—table 9 continued*

| Rank | Model | ELPD | ELPD Diff | SE of Diff | QQ RMSE |
|---|---|---|---|---|---|
| **Manicaland Male 35-39** | | | | | |
| 1 | Sinh-arcsinh | −11496.14 | 0.00 | 0.00 | 0.27 |
| 2 | Skew normal | −11528.36 | −32.22 | 9.83 | 0.39 |
| 3 | Normal | −11530.43 | −34.29 | 9.72 | 0.26 |
| 4 | Gamma | −11531.75 | −35.61 | 12.47 | 0.24 |
| 5 | Beta | −11582.63 | −86.49 | 12.97 | 0.48 |
| **Manicaland Male 40-44** | | | | | |
| 1 | Sinh-arcsinh | −8714.06 | 0.00 | 0.00 | 0.35 |
| 2 | Skew normal | −8749.78 | −35.72 | 10.11 | 0.51 |
| 3 | Gamma | −8777.08 | −63.02 | 10.38 | 0.55 |
| 4 | Normal | −8791.45 | −77.38 | 12.23 | 0.76 |
| 5 | Beta | −8860.22 | −146.16 | 18.15 | 0.93 |
| **Manicaland Male 45-49** | | | | | |
| 1 | Sinh-arcsinh | −7110.27 | 0.00 | 0.00 | 0.42 |
| 2 | Skew normal | −7177.03 | −66.75 | 25.08 | 0.76 |
| 3 | Gamma | −7213.99 | −103.72 | 13.02 | 1.07 |
| 4 | Normal | −7232.04 | −121.77 | 13.09 | 1.28 |
| 5 | Beta | −7312.35 | −202.08 | 18.61 | 1.61 |

**Appendix 1—table 10.** LOO-CV estimated ELPD values, differences, and standard errors of differences, as well as QQ RMSE values, for all five regression models fit to all four data sets.
The 'difference' value of a row is the difference between that row's ELPD value and data set-specific best ELPD value. Higher ELPD values and lower QQ RMSE values are better.

| Rank | Model | ELPD | ELPD Diff | SE of Diff | QQ RMSE |
|---|---|---|---|---|---|
| **AHRI** | | | | | |
| 1 | Distributional 4 | 55841.91 | 0.00 | 0.00 | 0.66 |
| 2 | Distributional 3 | 55534.16 | −307.75 | 32.36 | 0.93 |
| 3 | Distributional 2 | 54794.79 | −1047.12 | 51.69 | 1.21 |
| 4 | Distributional 1 | 54335.19 | −1506.72 | 72.32 | 1.15 |
| 5 | Conventional | 52689.21 | −3152.70 | 100.59 | 1.30 |
| **AHRI Deaheaped** | | | | | |
| 1 | Distributional 4 | 58503.98 | 0.00 | 0.00 | 0.62 |
| 2 | Distributional 3 | 58219.23 | −284.75 | 28.64 | 0.92 |
| 3 | Distributional 2 | 57503.68 | −1000.30 | 47.14 | 1.14 |
| 4 | Distributional 1 | 57097.39 | −1406.59 | 64.48 | 1.06 |
| 5 | Conventional | 55296.25 | −3207.73 | 99.42 | 1.26 |
| **Haiti 2016-17 DHS** | | | | | |
| 1 | Distributional 4 | 5207.57 | 0.00 | 0.00 | 0.84 |
| 2 | Distributional 3 | 5196.69 | −10.89 | 6.54 | 0.91 |
| 3 | Distributional 1 | 5140.77 | −66.80 | 12.27 | 0.98 |
| 4 | Distributional 2 | 5138.75 | −68.83 | 12.24 | 0.99 |
| 5 | Conventional | 4777.78 | −429.80 | 30.54 | 1.33 |

*Continued on next page*

*Appendix 1—table 10 continued*

| Rank | Model | ELPD | ELPD Diff | SE of Diff | QQ RMSE |
|------|-------|------|-----------|------------|---------|
| Manicaland | | | | | |
| 1 | Distributional 4 | 24516.15 | 0.00 | 0.00 | 1.04 |
| 2 | Distributional 3 | 24313.74 | −202.40 | 20.52 | 1.34 |
| 3 | Distributional 2 | 23472.07 | −1044.08 | 47.77 | 1.80 |
| 4 | Distributional 1 | 23192.49 | −1323.66 | 54.97 | 1.89 |
| 5 | Conventional | 21011.29 | −3504.86 | 89.01 | 2.05 |

