## [Decision Letter]

**Acceptance summary:**

This paper demonstrates the use of a sinh-arcsinh distribution to fit sex partner age distributions across different settings, and shows a superior fit compared to other more conventional distributions and regression methods. The methods outlined in this study will be of particular interest to social science researchers examining age mixing patterns, and for infectious disease modelers wishing to model sexual partner age mixing into their frameworks.

**Decision letter after peer review:**

Thank you for submitting your article "Evaluating distributional regression strategies for modelling self-reported sexual age-mixing" for consideration by *eLife*. Your article has been reviewed by 2 peer reviewers, one of whom is a member of our Board of Reviewing Editors, and the evaluation has been overseen by a Senior Editor. The following individual involved in review of your submission has agreed to reveal their identity: Adam Akullian (Reviewer #2).

As is customary in *eLife*, the reviewers have discussed their critiques with one another. What follows below is the Reviewing Editor's edited compilation of the essential and ancillary points provided by reviewers in their critiques and in their interaction post-review. Please submit a revised version that addresses these concerns directly.

Essential revisions:

1) Include more discussion about biases in self-reported data, including misreporting of partnership characteristics, selection bias, and missing data; also, discuss how these biases may affect the modelling of age mixing patterns.

2) Some methods could be further clarified by including text from the appendix into the main text.

3) Discuss how your results compare with and add to previous statistical frameworks for age mixing, notably log-linear models for mixing and Exponential-family Random Graph Models

*Reviewer #1:*

In this study, Wolock et al. compare the ability of different distributions to fit sexual age mixing patterns in a variety of settings, and provide a modeling framework for fitting these distributions across all ages without having to partition the data into separate datasets. They convincingly demonstrate that a sinh-arcsinh distribution will better fit the data than many other commonly used distributions, and also explore various ways of parameterizing age differences between partners. In general, the sinh-arcsinh produced both better fit statistics and more visually pleasing fits to the data. Moreover, the results confirm that more complex distributional assumptions with more parameters are in many cases warranted to fit age mixing patterns.

Particular strengths of this study are the exhaustive examination of multiple distributional assumptions, the use of multiple datasets to test the generalizability of results across regions, and use of objective fit statistics to discriminate between models and distributions. I do not see any major weaknesses, apart from maybe the issue of how possible it would be to fit these more complex distributions and models to smaller datasets, where there is less data to inform parameters.

As the authors point out, the methods outlined in this paper could be extended by including further stratification factors predicting age mixing, or potentially also mixing by some other continuous characteristics that are not expected to follow a normal distribution. The authors provided some sample code for fitting these distributions to the data, which should help others implement their proposed methods.

Comments for the authors:

• I had a lot of difficulty following the methods for the second aim, the regression component. I was very unclear on what was the outcome (dependent variables) being modeled and what were the parameters of the model. The appendix on model specifications helped to explain this a bit more. I think the manuscript would benefit from putting the model specification details from the appendix in the main text, as I could not understand what was being done without this further information.

• I find the heaping of ages surprising. This might have occurred due to the way the question about partner age was framed to participants. It might be useful to comment on the wording of this question across the different studies, and if there was a particular way of wording the question that led to more heaping in one study than in others.

• P. 8 I was initially confused as to why there are 36 subsets, given that there are only 2 sexes and 6 age groups between 20 and 50. It was not clear that there was a third stratification (survey). You should mention there are also 3 surveys (2*6*3=36).

• Table 1: the definition in the legend for Ce,d should probably read Ce,d(x).

*Reviewer #2:*

In "Evaluating distributional regression strategies for modelling self-reported sexual age-mixing" Wolock and colleagues present a statistical modeling framework to estimate the age-distribution of sexual partners from self-reported data. The authors use data from three population-level surveys where respondents report the ages of their most recent sexual partners. The authors use a Bayesian distributional regression approach to fit probability distribution parameters to the data and assess model performance. In this way, the authors demonstrate that a unique distribution of partnership ages can be reconstructed for each age and sex combination. The best fitting distributions and parameterizations are presented in the paper and the authors found that the sinh-arcsinh distribution most accurately reconstructed the distribution of partnership ages for each of the three geographies. Results from this analysis will be highly useful as a component of HIV (or other STI) risk prediction or to reconstruct sexual networks for mathematical models of STI transmission.

The main strength of this analysis is the simplicity of their modeling approach, which uses probability distributions with few parameters to capture the age distribution of sexual partnerships. This approach will allow researchers to easily simulate age- and sex-specific partnership probabilities and inherent uncertainty in those estimates without the need for complex empirical distributions.

Weaknesses of the analysis center around the inherent limitations of self-reported data on sexual partnership age. The authors use data on self-reported age(s) of a respondent's partnership(s) in the last 12 months. The authors don't consider potential reporting biases and missing data associated with self-report that may be differential by age, gender, and relationship type. For example, some individuals may selectively report certain types of partners over others – perhaps stable partnerships versus transient ones. Missing data is quite common for sensitive questions on sexual histories and thus the results presented here should be interpreted with that in mind. It is important to consider that some relationship types may be under-represented and may lead to biased age distributions.

Nonetheless, the results support the conclusions made by the authors. This paper makes an important methodological contribution to the literature on sexual networks and has broad application to understanding the transmission of STI's, especially HIV in diverse settings.

Overall this paper has a sound methodological approach and is clearly written.

1.Page 12, line 241, the number of partnerships is indicated for each geography but not the number of respondents. Would be good to understand how many individuals reported just one partner versus more than one. Also, I assume the same partner could be reported multiple times at different survey rounds for the two longitudinal datasets, which could create pseudo replication. Could the authors discuss how this type of correlated data might influence the uncertainty estimates?

*Reviewer #3:*

The primary weakness with this manuscript is that the authors appear to be unaware of two robust statistical frameworks that have been used to estimate age mixing (and other forms of heterogeneous mixing) for many years.

The first is the use of log-linear models for mixing matrices (Morris, 1991, 1993). These models can be used to fit a wide array of mixing patterns, both discrete and continuous, and are also quite general. They offer everything from parsimonious summary parameterization (with assessments of goodness of fit) to dummy parameterization for each cell in the matrix to get exact fits. They benefit from the well developed statistical theory and methods that support generalized linear models. While they can be generalized to include random effects for persons, the limitations of these models is that they can not be used to represent "edge dependence" (think monogamy bias – the presence of one spouse ("edge") has a profound effect on the probabilty of another spouse for the same individual). The papers above integrate the resulting estimates into a standard deterministic compartmental model for epidemic simulation.

The second is the more recent Exponential-family Random Graph Models (ERGMs, and their temporal cousins TERGMs – references provided below). These models are also a form of GLM, but extend the log-linear framework with the capability to represent dependent edges. They are designed primarily for use with stochastic network epidemic models, and are sufficiently well established that they serve as the foundation for the EpiModel (Jenness et al. 2018) software package.

In both cases, the benefit of these models is that they are a general, principled statistical framework, with well understood properties and established methods for model testing and evaluation. This enables the analyst to model not only age mixing, but other forms of heterogeneous mixing (e.g., by sex, or geographic location, or race/ethnicity), and, in the case of TERGMs, degree distributions and higher order network configurations, along with heterogeneous edge duration. All of these terms can be estimated jointly, from sampled network data, and simulations from the estimated model will reproduce the sufficient statistics in expectation. This latter property makes these models a natural foundation for epidemic simulation.

As a result, the added value of this distributional regression strategy is unclear. And this statement: "More broadly, no previous work has systematically evaluated the wide variety of distributions potentially available to model partner age distributions" is simply untrue. Both frameworks above provide general methods for fitting and assessing a wide variety of distributions.

References:

Morris, M. (1991). "A log-linear modeling framework for selective mixing." Math Biosc 107: 349-377.

Morris, M. (1993). "Epidemiology and Social Networks: Modeling Structured Diffusion." Soc. Meth. Res. 22(1): 99-126.

Jenness, S. M., S. M. Goodreau and M. Morris (2018). "EpiModel: An R Package for Mathematical Modeling of Infectious Disease over Networks." Journal of Statistical Software 84(8).

Selected refs for ERGMs and TERGMs

Hunter, D. R., M. S. Handcock, C. T. Butts, S. M. Goodreau and M. Morris (2008). "ergm: A Package to Fit, Simulate and Diagnose Exponential-Family Models for Networks." Journal of Statistical Software 42(i03).

Hunter, D. R., S. M. Goodreau and M. S. Handcock (2008). "Goodness of fit of social network models." Journal of the American Statistical Association 103(481): 248-258.

Krivitsky, P. N. and M. S. Handcock (2014). "A separable model for dynamic networks." Journal of the Royal Statistical Society, Series B 76(1): 29-46.

Krivitsky, P. N., M. S. Handcock and M. Morris (2011). "Adjusting for Network Size and Composition Effects in Exponential-Family Random Graph Models." Statistical Methodology 8(4): 319–339.

Krivitsky, P. N. and M. Morris (2017). "Inference for social network models from egocentrically sampled data, with application to understanding persistent racial disparities in HIV prevalence in the US." Annals of Applied Statistics 11(1): 427-455.

---

## [Author Response]

Essential revisions:1) Include more discussion about biases in self-reported data, including misreporting of partnership characteristics, selection bias, and missing data; also, discuss how these biases may affect the modelling of age mixing patterns.

Thank you for this suggestion. The reviewers raise a good point about missingness and bias. Our method cannot resolve the limitations inherent in self-reported data, but, for certain types of missingness, it could be used to smooth and interpolate missing data. We have included a paragraph (lines 456-463) addressing this limitation:

“Finally, our model does not address any reporting biases in self-reported partnership data. If certain relationship types are perceived as less socially acceptable, respondents might be less likely to report them, resulting in systematic missingness. Our method could still be appropriate to model the age distribution from the data reported about an under-reported partnership type, but it cannot predict whether or not a given partnership exists. However, if under-reporting correlates with partner age (or age difference), then the empirical distributions will be biased and our method will only smooth and interpolate the biased data.”

2) Some methods could be further clarified by including text from the appendix into the main text.

Thank you for this feedback. We have moved some of the explanation of the distributional regression methods from the Appendix back into the main text. In lines 235-240 we have included a general description of the models we fit, and in lines 241-245, we have clarified that our five regression models vary in how their design matrices are defined:

“By varying the specifications of the four design matrices, X^µ^, X^σ^, X^ε^, and X^δ^, we tested how well a series of increasingly complex distributional models fit to each data set. We fit the following models, which varied in the definitions of their four design matrices:”

3) Discuss how your results compare with and add to previous statistical frameworks for age mixing, notably log-linear models for mixing and Exponential-family Random Graph Models

Thank you for this suggestion. We focused our analysis on a relatively narrow problem of identifying continuous parametric models for sexual partner ages conditional on an individual’s age and sex. We agree with the editors and reviewer that it is valuable to connect our work to the broader context of more general and flexible statistical approaches to model sexual partner mixing matrices and sexual network, and greatly appreciate the suggestion.

We have elaborated the introductory paragraph on previous approaches to modelling sexual partner age distributions to also draw connections to this wider literature (lines 52-77; revisions emphasised below):

“One may consider statistical modelling approaches for the distribution of partner age as a function of respondent age and sex. […] These stochastic methods, along with the broader suite of ERGMs (David R. Hunter et al. 2008a; David R Hunter et al. 2008b; Krivitsky and Handcock 2014; Krivitsky, Handcock, and Martina Morris 2011), can model social network data accurately with robust incorporation of covariates, and tools exist to incorporate their estimates into epidemic models (Morris 1993; Jenness et al. 2018).”

In the penultimate paragraph of the introduction (line 78), we have clarified the focus of our analysis in the context of this wider research area:

“We focused on evaluating parametric models for continuously representing the distribution of sexual partner ages conditional on the respondent's age…”

Finally in the Discussion section (line 481), we have noted the potential opportunity to use sinh-arcsinh distribution within more comprehensive models for mixing matrices, such as those described above:

“The distribution regression framework with the sinh-arcsinh may also be a useful parametric model for continuous representation of marginal distributions within sexual mixing models or network models, such as the ERGM framework.”

Reviewer #1:In this study, Wolock et al. compare the ability of different distributions to fit sexual age mixing patterns in a variety of settings, and provide a modeling framework for fitting these distributions across all ages without having to partition the data into separate datasets. They convincingly demonstrate that a sinh-arcsinh distribution will better fit the data than many other commonly used distributions, and also explore various ways of parameterizing age differences between partners. In general, the sinh-arcsinh produced both better fit statistics and more visually pleasing fits to the data. Moreover, the results confirm that more complex distributional assumptions with more parameters are in many cases warranted to fit age mixing patterns.Particular strengths of this study are the exhaustive examination of multiple distributional assumptions, the use of multiple datasets to test the generalizability of results across regions, and use of objective fit statistics to discriminate between models and distributions. I do not see any major weaknesses, apart from maybe the issue of how possible it would be to fit these more complex distributions and models to smaller datasets, where there is less data to inform parameters.As the authors point out, the methods outlined in this paper could be extended by including further stratification factors predicting age mixing, or potentially also mixing by some other continuous characteristics that are not expected to follow a normal distribution. The authors provided some sample code for fitting these distributions to the data, which should help others implement their proposed methods.Comments for the authors:• I had a lot of difficulty following the methods for the second aim, the regression component. I was very unclear on what was the outcome (dependent variables) being modeled and what were the parameters of the model. The appendix on model specifications helped to explain this a bit more. I think the manuscript would benefit from putting the model specification details from the appendix in the main text, as I could not understand what was being done without this further information.

Thank you for raising this point. As discussed in our second response to the *Essential Revisions* above, we have provided a specific definition of our sinh-arcsinh distributional model in the main text (including a definition of the dependent variable) in lines 235-240. In lines 241-245, we have clarified that our five distributional models vary in their definitions of the four design matrices. We hope that these changes clarify the methods.

• I find the heaping of ages surprising. This might have occurred due to the way the question about partner age was framed to participants. It might be useful to comment on the wording of this question across the different studies, and if there was a particular way of wording the question that led to more heaping in one study than in others.

The reviewer is correct that heaping is determined how the questions are phrased. The AHRI survey asks how much older/younger the partner is, so respondents round age differences to multiples of five (as opposed to rounding partner ages themselves). We have included a description of this phenomenon in the main text in lines 151-154: “For example, if a questionnaire asks "how many years older or younger is your partner than you?", respondents might be disproportionately likely to report a multiple of five, leading to age differences that are heaped on multiples of five.”

• P. 8 I was initially confused as to why there are 36 subsets, given that there are only 2 sexes and 6 age groups between 20 and 50. It was not clear that there was a third stratification (survey). You should mention there are also 3 surveys (2*6*3=36).

Thank you for this feedback. We have clarified that the data subsets are data set-, sex-, and age bin-specific in line 214: replaced “all 36 subsets” with “all 36 data set-/sex-/age bin-specific subsets”. In line 160 we have replaced “each data set” with “all three data sets”

• Table 1: the definition in the legend for C_ε,δ_ should probably read C_ε,δ_(x).

Thank you for pointing out this mistake. We have replaced “C_ε,δ_” with “C_ε,δ_(x)” in the caption of Table 1.

Reviewer #2:[…] Overall this paper has a sound methodological approach and is clearly written. The following are suggestions to improve clarity and understanding:1.Ppage 12, line 241, the number of partnerships is indicated for each geography but not the number of respondents. Would be good to understand how many individuals reported just one partner versus more than one. Also, I assume the same partner could be reported multiple times at different survey rounds for the two longitudinal datasets, which could create pseudo replication. Could the authors discuss how this type of correlated data might influence the uncertainty estimates?

The reviewer raises an excellent point that we did not address in the original submission. We view both of these forms of structure in the data sets as violations of the independence we have assumed across observations. Because of this pseudo-replication, we suspect that we are attributing too much information to respondents who are repeated in the data and therefore might be deflating estimated uncertainty. We have included the number of individual respondents and the resulting average partners per respondent in lines 292-285: “There were 36,033 respondents reporting at least one partnership in the AHRI data, 25,024 in the Manicaland data, and 12,143 in the Haiti DHS, resulting in averages of 2.2, 2.3, and 1.0 partners per respondent, respectively.”

We have also included a paragraph (lines 447-455) identifying both of these details as a limitation of the method:

“There are two sources of possible non-independence in our data sets that were not modelled. […] Because we modelled multiple observations of the same individual as conditionally independent, we anticipate that these correlations may artificially increase the precision of our estimates.”

Reviewer #3:The primary weakness with this manuscript is that the authors appear to be unaware of two robust statistical frameworks that have been used to estimate age mixing (and other forms of heterogeneous mixing) for many years.The first is the use of log-linear models for mixing matrices (Morris, 1991, 1993). These models can be used to fit a wide array of mixing patterns, both discrete and continuous, and are also quite general. They offer everything from parsimonious summary parameterization (with assessments of goodness of fit) to dummy parameterization for each cell in the matrix to get exact fits. They benefit from the well developed statistical theory and methods that support generalized linear models. While they can be generalized to include random effects for persons, the limitations of these models is that they can not be used to represent "edge dependence" (think monogamy bias – the presence of one spouse ("edge") has a profound effect on the probabilty of another spouse for the same individual). The papers above integrate the resulting estimates into a standard deterministic compartmental model for epidemic simulation.The second is the more recent Exponential-family Random Graph Models (ERGMs, and their temporal cousins TERGMs – references provided below). These models are also a form of GLM, but extend the log-linear framework with the capability to represent dependent edges. They are designed primarily for use with stochastic network epidemic models, and are sufficiently well established that they serve as the foundation for the EpiModel (Jenness et al. 2018) software package.In both cases, the benefit of these models is that they are a general, principled statistical framework, with well understood properties and established methods for model testing and evaluation. This enables the analyst to model not only age mixing, but other forms of heterogeneous mixing (e.g., by sex, or geographic location, or race/ethnicity), and, in the case of TERGMs, degree distributions and higher order network configurations, along with heterogeneous edge duration. All of these terms can be estimated jointly, from sampled network data, and simulations from the estimated model will reproduce the sufficient statistics in expectation. This latter property makes these models a natural foundation for epidemic simulation.As a result, the added value of this distributional regression strategy is unclear. And this statement: "More broadly, no previous work has systematically evaluated the wide variety of distributions potentially available to model partner age distributions" is simply untrue. Both frameworks above provide general methods for fitting and assessing a wide variety of distributions.References:Morris, M. (1991). "A log-linear modeling framework for selective mixing." Math Biosc 107: 349-377.Morris, M. (1993). "Epidemiology and Social Networks: Modeling Structured Diffusion." Soc. Meth. Res. 22(1): 99-126.Jenness, S. M., S. M. Goodreau and M. Morris (2018). "EpiModel: An R Package for Mathematical Modeling of Infectious Disease over Networks." Journal of Statistical Software 84(8).Selected refs for ERGMs and TERGMsHunter, D. R., M. S. Handcock, C. T. Butts, S. M. Goodreau and M. Morris (2008). "ergm: A Package to Fit, Simulate and Diagnose Exponential-Family Models for Networks." Journal of Statistical Software 42(i03).Hunter, D. R., S. M. Goodreau and M. S. Handcock (2008). "Goodness of fit of social network models." Journal of the American Statistical Association 103(481): 248-258.Krivitsky, P. N. and M. S. Handcock (2014). "A separable model for dynamic networks." Journal of the Royal Statistical Society, Series B 76(1): 29-46.Krivitsky, P. N., M. S. Handcock and M. Morris (2011). "Adjusting for Network Size and Composition Effects in Exponential-Family Random Graph Models." Statistical Methodology 8(4): 319–339.Krivitsky, P. N. and M. Morris (2017). "Inference for social network models from egocentrically sampled data, with application to understanding persistent racial disparities in HIV prevalence in the US." Annals of Applied Statistics 11(1): 427-455.

We thank the Reviewer for these suggestions. As we have indicated in our responses to the Essential Revisions above, we conceptualised the focus of our work on the somewhat narrower problem of continuously modelling the distributional features of sexual partner age mixing. We agree with, and greatly appreciate, the suggestion to draw connection and situate our work in the context of these powerful and flexible statistical approaches to modelling mixing matrices and sexual networks. Our specific revisions, including references to the literature highlighted above, are detailed in our responses to the ‘Essential revisions’ section above.